# Ciliary and extraciliary Gpr161 pools repress hedgehog signaling in a tissue-specific manner

Sun-Hee Hwang, Bandarigoda N Somatilaka[†], Kevin White, Saikat Mukhopadhyay*

Department of Cell Biology, University of Texas Southwestern Medical Center, Dallas, United States

*For correspondence:
saikat.mukhopadhyay@
utsouthwestern.edu

Present address: [†]Department of Dermatology, University of Texas Southwestern Medical Center, Dallas, United States

Competing interests: The authors declare that no competing interests exist.

**Abstract** The role of compartmentalized signaling in primary cilia during tissue morphogenesis is not well understood. The cilia localized G protein-coupled receptor, Gpr161, represses hedgehog pathway via cAMP signaling. We engineered a knock-in at the *Gpr161* locus in mice to generate a variant (Gpr161[mut1]), which was ciliary localization defective but cAMP signaling competent. Tissue phenotypes from hedgehog signaling depend on downstream bifunctional Gli transcriptional factors functioning as activators or repressors. Compared to knockout (ko), *Gpr161[mut1/ko]* had delayed embryonic lethality, moderately increased hedgehog targets, and partially down-regulated Gli3 repressor. Unlike ko, the *Gpr161[mut1/ko]* neural tube did not show Gli2 activator-dependent expansion of ventral-most progenitors. Instead, the intermediate neural tube showed progenitor expansion that depends on loss of Gli3 repressor. Increased extraciliary receptor levels in *Gpr161[mut1/mut1]* prevented ventralization. Morphogenesis in limb buds and midface requires Gli repressor; these tissues in *Gpr161[mut1/mut1]* manifested hedgehog hyperactivation phenotypes—polydactyly and midfacial widening. Thus, ciliary and extraciliary Gpr161 pools likely establish tissue-specific Gli repressor thresholds in determining morpho-phenotypic outcomes.

## Introduction

The primary cilium is a paradigmatic organelle for studying compartmentalized cellular signaling in morphogenesis and disease (*Anvarian et al., 2019*; *Nachury and Mick, 2019*). The primary cilium is a microtubule-based dynamic cellular appendage that is templated from the mother centriole of the centrosome–the basal body, and is present in multiple cell types (*Wang and Dynlacht, 2018*). Cilia can transduce cellular response to extracellular signals, such as hedgehog (Hh) ligands (*Huangfu et al., 2003*), during differentiation and proliferation, regulating morphogenesis (*Kopinke et al., 2021*). Long considered vestigial, dysfunction of the primary cilium has now been implicated in diseases affecting diverse tissues, collectively termed 'ciliopathies' (*Reiter and Leroux, 2017*). The tissue abnormalities include neural tube defects (*Murdoch and Copp, 2010*), limb/skeletal malformations (*Huber and Cormier-Daire, 2012*), craniofacial dysmorphisms (*Brugmann et al., 2010b*), and heart defects (*Gabriel et al., 2021*), emphasizing ciliary role in diverse contexts.

The mechanisms by which cilia-specific signals are maintained and propagated to direct downstream pathways during morphogenesis is not well understood. Particularly, non-ciliary signaling by cilia localized proteins can be a confounding factor in understanding how signaling at and by cilia contributes to tissue phenotypes. Studying signaling at cilia requires mechanistic understanding of trafficking to cilia, isolating ciliary from extraciliary functions of signaling molecules, and studying functional consequences directly in tissues without disrupting cilia (*Mukhopadhyay et al., 2017*). Decoding the features of signaling unique to the primary cilium in directing downstream cellular pathways is necessary for understanding the pathogenesis of diseases caused by ciliary dysfunction.

The primary cilium mediates cellular signaling responses to Hh morphogens in vertebrates (*Briscoe and Thérond, 2013*; *Goetz and Anderson, 2010*). An intricate balance between formation of Gli transcriptional activators and repressors determines the transcriptional output to Hh morphogens (*Hui and Angers, 2011*). Both responses are dependent on the primary cilium. Binding of Hh to its receptor Patched (Ptch1) triggers removal of Ptch1 from cilia and promotes enrichment and activation of Smoothened (Smo)—the pathway transducer—in cilia, resulting in generation of Gli transcriptional activator (GliA) (*Corbit et al., 2005*; *Rohatgi et al., 2007*). In contrast, repression in the absence of Hh involves protein kinase A (PKA) initiated phosphorylation followed by limited proteolysis of full-length Gli2/3 into Gli2/3 repressor (GliR) forms, also in a cilia-dependent manner (*Tuson et al., 2011*; *Mukhopadhyay and Rohatgi, 2014*). GliA/R regulation is also different in tissues and can be categorized as primarily occurring by GliA or GliR thresholds (*Kopinke et al., 2021*).

We previously described that the cilia-localized orphan G-protein-coupled receptor (GPCR), Gpr161 functions as a negative regulator of Hh signaling during early neural tube development in mice (*Mukhopadhyay et al., 2013*). Mice knockout for *Gpr161* are embryonic lethal by embryonic day 10.5 (E10.5) and exhibit increased Hh signaling and expansion of ventral progenitors throughout the rostrocaudal extent of the neural tube, without disrupting cilia. Gpr161 determines Gli3R formation via cAMP-PKA signaling (*Bachmann et al., 2016*; *Mukhopadhyay et al., 2013 Tschaikner et al., 2021*). However, whereas ventral neural tube is mainly regulated by Gli2 and Gli3 activators (*Bai and Joyner, 2001*; *Bai et al., 2004*), the intermediate region of the neural tube is regulated by Gli3 repressor (*Persson et al., 2002*). We recently also described roles of Gpr161 in forelimb formation, skeletal development (*Hwang et al., 2018*), embryonic cerebellar development (*Shimada et al., 2018*), and forebrain development (*Shimada et al., 2019*). Tissues such as the limbbuds and cerebellum show GliR gradients preceding sonic hedgehog (Shh) expression (*Kopinke et al., 2021*), where *Gpr161* deletion resulted in lack of GliR (*Hwang et al., 2018*; *Shimada et al., 2018*). Thus, *Gpr161* deletion can induce high Hh signaling phenotypes in tissues that are regulated by loss of GliR and/or from excessive GliA formation.

Gpr161 localizes to the ciliary membrane dynamically and also to the recycling endocytic compartment (*Mukhopadhyay et al., 2013*; *Pal et al., 2016*). However, whether Gpr161 functions from inside the primary cilium and/or in the endomembrane compartment in regulating cAMP-PKA signaling and Hh pathway phenotypes during morphogenesis is not clear. Importantly, the phenotypes in all affected tissues from *Gpr161* deletion were rescued from concomitant loss of cilia (*Hwang et al., 2018*; *Mukhopadhyay et al., 2013*; *Shimada et al., 2018*) indicating that the consequences from lack of Gpr161 signaling were strictly cilia dependent. Interestingly, Gpr161 C-tail has been proposed to be an A-kinase anchoring protein (AKAP) for PKA activation in cilia by binding to type I PKA regulatory subunits (*Bachmann et al., 2016*). The PKA regulatory subunit RIα and RII localizes to cilia (*Bachmann et al., 2016*; *Mick et al., 2015*) and centrosomes (*Barzi et al., 2010*; *Saade et al., 2017*; *Tuson et al., 2011*), respectively. PKA-c also localizes to centrosomes (*Barzi et al., 2010*; *Saade et al., 2017*; *Tuson et al., 2011*) and cilia (*Arveseth et al., 2021*; *Truong et al., 2021*). Gpr161 is the only GPCR known to be an AKAP. Precise regulation of Gpr161 pools inside and/or outside cilia is required to address role of subcellular compartmentalization of cAMP in morpho-phenotypic outcomes.

Here, we engineered a knock-in at the endogenous *Gpr161* locus in mice to generate a variant (Gpr161^mut1) that is ciliary localization defective but signaling competent. We demonstrated the tubby family protein Tulp3 (*Badgandi et al., 2017*; *Mukhopadhyay et al., 2010*) as a key adapter that traffics multiple types of cargoes into ciliary membrane without affecting total cargo pools. The *Gpr161* knock-in mutant was based on the Tulp3-targeted ciliary localization sequence that we had previously identified (*Mukhopadhyay et al., 2013*). Using this knock-in model, we demonstrate that tissues that require GliR for morphogenesis, such as mid face and limb buds, are affected from loss of Gpr161 ciliary pools showing mid face widening and polydactyly. In the neural tube, intermediate level patterning that is regulated by Gli3R is affected in *Gpr161^mut1/ko* embryos, but ventral-most patterning that is GliA regulated remains unaffected. Our findings suggest that ciliary Gpr161 pools prevent Hh pathway hyperactivation phenotypes likely from GliR lack but not from GliA generation.

## Results

### A ciliary localization-defective *Gpr161* mutant is signaling-competent

We previously described the (V/I)KARK motif in the third intracellular loop of Gpr161 to be necessary and sufficient for targeting to cilia by the tubby family protein Tulp3 (*Figure 1A*; *Badgandi et al., 2017*; *Mukhopadhyay et al., 2010*; *Mukhopadhyay et al., 2013*). We used NIH 3T3 cells that are knockout for *Gpr161* (*Pusapati et al., 2018*) to stably overexpress untagged wildtype *Gpr161* and *Gpr161^mut1* (VKARK>AAAAA; Gpr161^mut1). Wildtype Gpr161 predominantly localized to cilia. In contrast, Gpr161^mut1 was not localized to cilia but was observed in vesicles, some of which surrounded the base of cilia (*Figure 1B*).

We previously showed that Tulp3 regulates trafficking of Gpr161 into cilia but does not affect trafficking out of cilia (*Badgandi et al., 2017*). We used three approaches in cells stably expressing LAP-tagged Gpr161^mut1 (LAP, S tag-PreScission-GFP) to promote Gpr161 accumulation in cilia or prevent Gpr161 trafficking out of cilia and detect Gpr161^mut1 by immunostaining for GFP. These experiments conclusively demonstrate that Gpr161^mut1 does not at all transit through cilia.

First, loss of the 5'phosphatase, Inpp5e, increases levels of Tulp3, intraflagellar transport complex-A (IFT-A) subunits and Gpr161 in the ciliary membrane (*Chávez et al., 2015*; *Garcia-Gonzalo et al., 2015*) likely from defects in Tulp3 cargo release (*Badgandi et al., 2017*). If Gpr161^mut1 transits through cilia but is too low to detect at steady state, we would expect to see it accumulating upon Inpp5e loss. By stably expressing the mutant LAP-tagged wild type (*Gpr161^wt*) and *Gpr161^mut1* in RPE hTERT cells, we confirmed the *Gpr161^mut1* fusion to be not localizing to cilia unlike *Gpr161^wt* (*Figure 1C–D*). As expected, upon *INPP5E* knockdown, we saw accumulation of endogenous TULP3 and LAP-tagged Gpr161^wt in cilia (*Figure 1C–F*). However, we did not see any accumulation of LAP-tagged *Gpr161^Mut1* in cilia (*Figure 1C–E*), despite accumulation of TULP3 (*Figure 1F*).

Second, Gpr161 is accumulated in β−arrestin 1/2 double knockout (*Arrb1/Arrb2* dko) MEFs as β−arrestins regulate Gpr161 removal from cilia (*Pal et al., 2016*). If Gpr161^mut1 transits through cilia, lack of β−arrestins should prevent removal from cilia promoting ciliary accumulation. We stably expressed LAP-tagged *Gpr161^wt* and *Gpr161^mut1* in wild type and *Arrb1/Arrb2* dko MEFs (*Kovacs et al., 2008*; *Figure 1—figure supplement 1*). LAP-tagged Gpr161^wt localized to cilia and was removed upon treatment with Smoothened agonist SAG but was accumulated in dko cells, as reported earlier (*Pal et al., 2016*). However, LAP-tagged Gpr161^mut1 did not accumulate in cilia of wild type or dko MEFs. Accumulation of endogenous Gpr161 in the *Arrb1/Arrb2* dko MEFs expressing LAP-tagged Gpr161^mut1, surmised from Gpr161 immunostaining in absence of any GFP in cilia, was still observed.

Third, the BBS-ome has been shown to regulate exit of Gpr161 from cilia (*Ye et al., 2018*) and knockdown of BBS-ome subunits result in accumulation of Gpr161 in cilia (*Nozaki et al., 2018*). If Gpr161^mut1 transits through cilia, lack of BBS subunits should prevent exit from cilia promoting ciliary accumulation. Stably expressed LAP-tagged Gpr161^wt localized to cilia and was accumulated upon *BBS4* knockdown in RPE-hTERT cells. However, stably expressed LAP-tagged Gpr161^mut1 did not accumulate in cilia even upon *BBS4* knockdown (*Figure 1—figure supplement 2*).

We had previously demonstrated wild type Gpr161 containing vesicles to be recycling endosomes that co-label with endocytosed fluorescent transferrin (*Mukhopadhyay et al., 2013*). We noted that LAP-tagged Gpr161^mut1 containing vesicles co-labeled with endocytosed fluorescent transferrin, similar to LAP-tagged Gpr161^wt (*Figure 1—figure supplement 3*). Thus, the propensity of the receptor to reside in recycling endosomes is not affected even in the absence of transit of Gpr161^mut1 through cilia.

Agonist-independent constitutive signaling has been observed for a wide variety of GPCRs (*Leurs et al., 1998*; *Seifert and Wenzel-Seifert, 2002*). In the absence of a known ligand for the orphan GPCR, Gpr161, we previously demonstrated constitutive activity of the receptor by generating doxycycline-inducible stable cell lines expressing the untagged wild-type receptor (*Mukhopadhyay et al., 2013*). We used a similar strategy to generate a doxycycline-inducible line expressing untagged *Gpr161^mut1*. We used a TR-FRET cAMP assay (Cisbio) for estimating cAMP levels. These assays eliminate the background noise due to FRET-based estimation of signals, and the use of long-lived fluorophores (in this case lathanides), combined with time-resolved FRET detection (a delay between excitation and emission detection) minimizes prompt intrinsic

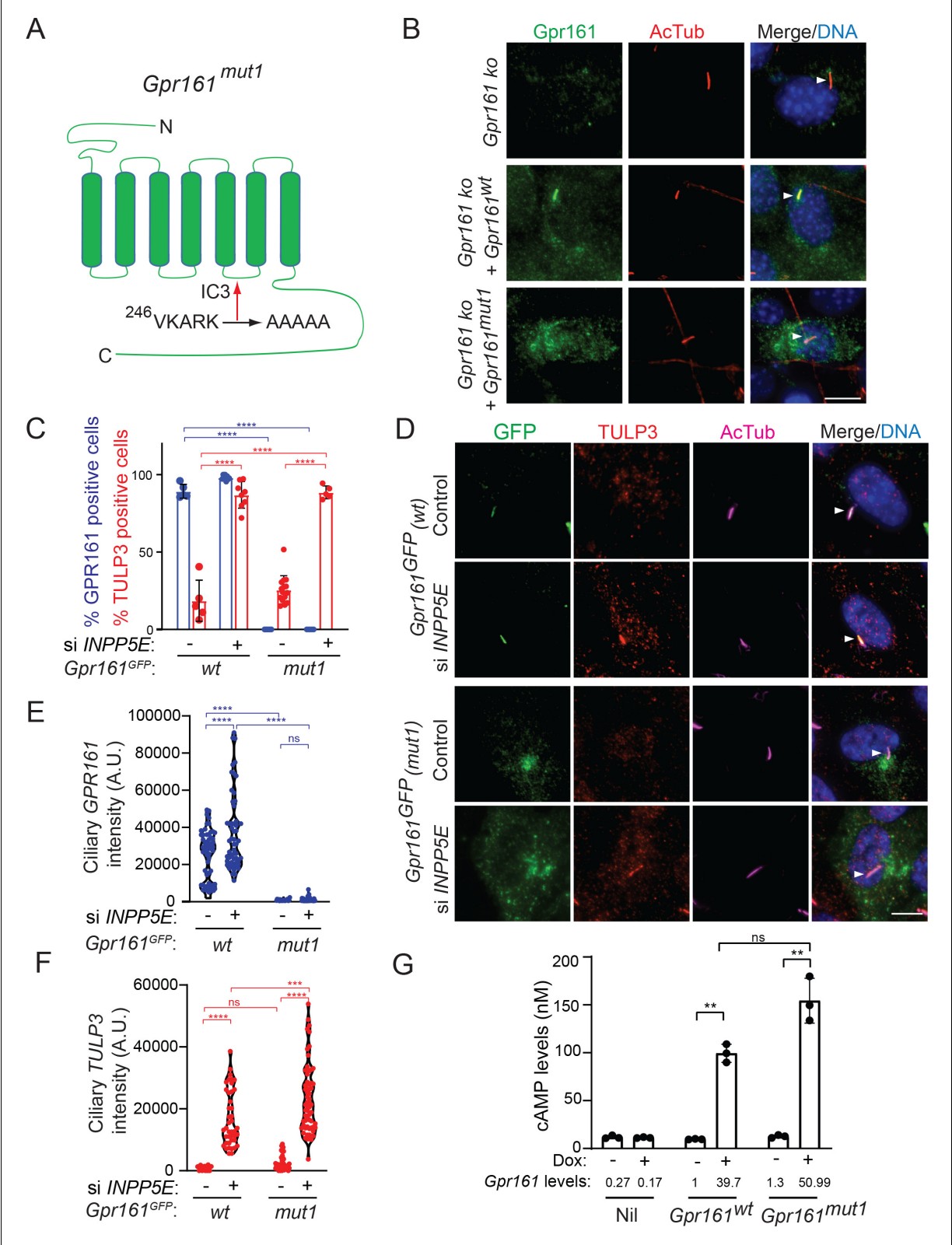

**Figure 1.** A ciliary localization defective *Gpr161* mutant is competent for cAMP signaling. (**A**) Cartoon representing the VKARK>AAAAA mut1 mutation in the third intracellular loop of mouse Gpr161. (**B**) Confluent NIH 3T3 Flp-In CRISPR-based *Gpr161* knockout (ko) cells stably expressing untagged wildtype (wt) or Gpr161^mut1 were starved for 24 hr, fixed, and immunostained with anti-Gpr161 (green), anti-acetylated tubulin (AcTub; red) antibodies and counterstained for DNA (blue). Arrowheads indicate cilia. (**C – D**) RPE hTERT cells stably expressing C-terminal LAP-tagged *Gpr161^wt* or *Gpr161^mut1*

*Figure 1 continued on next page*

*Figure 1 continued*

constructs were sequentially transfected with control or *INPP5E* siRNA (100 nM) twice and cultured for a total of 72 hr. The cells were serum starved for the last 24 hr before fixation and immunostained with anti-GFP (green), anti-TULP3 (red), anti-acetylated tubulin (AcTub; magenta) antibodies and counterstained for DNA (blue). GPR161 and TULP3 positive cells were quantified. Arrowheads in (**D**) indicate cilia. Total 6–12 different images quantified from two experiments, and total 600–2000 cells counted/condition. Data shown as mean ± SD. ****, p<0.0001. Other pairwise comparisons are not significantly different. (**E – F**) Quantification of LAP-tagged Gpr161 (**E**) and TULP3 (**F**) pixel intensities from C shown as violin plots. Total counted cells were >60/condition. A.U., arbitrary units. ****, p<0.0001; ***, p<0.001. (**G**) Doxycycline-inducible NIH 3T3 Tet-on 3G cells expressing untagged *Gpr161*^wt^ or *Gpr161*^mut1^ were induced for 24 hr with 2 μg/ml doxycycline. The cells were subjected to TR-FRET cell-based assays for assaying cAMP. cAMP levels (nM) were calculated as interpolated values from a standard curve. Data from triplicate wells (mean ± SD) and is representative of 3 independent experiments. Mean *Gpr161* transcript levels are shown below. **, p<0.01. ns, not significant. Scale: (**B** and **D**), 10 μm.

The online version of this article includes the following source data and figure supplement(s) for figure 1:

**Source data 1.** Table showing data from experiments plotted in *Figure 1C*.
**Source data 2.** Table showing cilia intensity data from plotted in *Figure 1E and F*.
**Source data 3.** Table showing cAMP data plotted in *Figure 1G*.
**Figure supplement 1.** Gpr161^mut1^ was not accumulated in cilia of β−arrestin 1/2 double knockout MEFs.
**Figure supplement 2.** Gpr161^mut1^ was not accumulated in cilia upon *BBS4* knockdown in RPE-hTERT cells.
**Figure supplement 3.** Gpr161^mut1^ containing vesicles co-label with endocytosed transferrin similar to Gpr161^wt^.

fluorescence interferences (*Degorce et al., 2009*). Gpr161^mut1^ demonstrated comparable constitutive cAMP signaling activity compared to the wild-type receptor (*Figure 1G*), suggesting that the ciliary localization defective GPCR was signaling competent. Thus, the *Gpr161*^mut1^ mutant allows to uncouple cAMP signaling function from ciliary localization of Gpr161.

## Generating ciliary localization defective endogenous knock-in *Gpr161*^mut1^ mouse model

We next generated a mouse knock-in allele of *Gpr161* (*Gpr161*^mut1^) in the endogenous locus and confirmed the endogenous integration by southern blotting (*Figure 2A–B*) and sequencing (*Figure 2C*). We engineered a NotI restriction site in the mutated *Gpr161*^mut1^ exon four sequence (*Figure 2—figure supplement 1*) for genotyping using RFLP assays (*Figure 2A and D*). We bred the knock-in alleles with *Gpr161* knockout (ko) allele. Quantitative RT-PCR of *Gpr161* transcripts in E9.5 embryos suggested that the *Gpr16*^mut1^ transcript was expressed at similar levels compared to wild type and was reduced to 50% compared to wildtype in *Gpr161*^ko/mut1^ embryos (*Figure 2E*). *Gpr161*^ko/ko^ embryos lacked *Gpr161* transcripts, as expected.

We were unable to determine protein levels of the endogenous mutant receptor in the *Gpr161*^mut1^ embryos due to technical constraints in immunoblotting for endogenous levels. However, we note Gpr161^mut1^ in vesicles surrounding the base of cilia (*Figure 1B*) and constitutive cAMP signaling activity (*Figure 1G*) in stable cell lines, suggesting that protein levels and activity of the mutant were comparable with wild-type Gpr161. We stably overexpressed LAP-tagged Gpr161^wt^ and Gpr161^mut1^ in wild-type MEFs and performed tandem affinity purification followed by immunoblotting to detect the proteins. We noted similar immunoblotting pattern from receptor glycosylation in both variants (*Figure 2—figure supplement 2*).

We previously reported that the morphology of cilia was not grossly affected in the *Gpr161* knockout or conditional knockout embryos (*Hwang et al., 2018*; *Mukhopadhyay et al., 2013*). The ciliary morphologies, including ciliary lengths in *Gpr161*^ko/mut1^ embryos in mesenchyme, somatopleuric mesoderm or in the neural tube were similarly unaffected (*Figure 2—figure supplement 3*).

## *Gpr161*^mut1^ allele is hypomorphic to knockout

Mice homozygous for *Gpr161* ko allele are embryonic lethal by E10.5 with extensive craniofacial abnormalities, open forebrain and midbrain regions, and lack of forelimbs (*Hwang et al., 2018*; *Mukhopadhyay et al., 2013*; *Figure 3A*). In contrast, the *Gpr161*^mut1/ko^ mice were embryonic lethal by E13.5 (*Table 1*). The *Gpr161*^mut1/ko^ embryos had craniofacial abnormalities, microphthalmia (*Figure 3A–C*) and spina bifida similar to *Gpr161* ko (*Figure 3B*) along with kinked tail, often associated with caudal neural tube defects (*Figure 3C*). They also had stunted forelimbs (*Figure 3B–C*) and heart defects with pericardial effusion (*Figure 3C*). However, the *Gpr161*^mut1/mut1^ mice were embryonic lethal by E14.75 (*Table 1*) with craniofacial abnormalities including mid face widening,

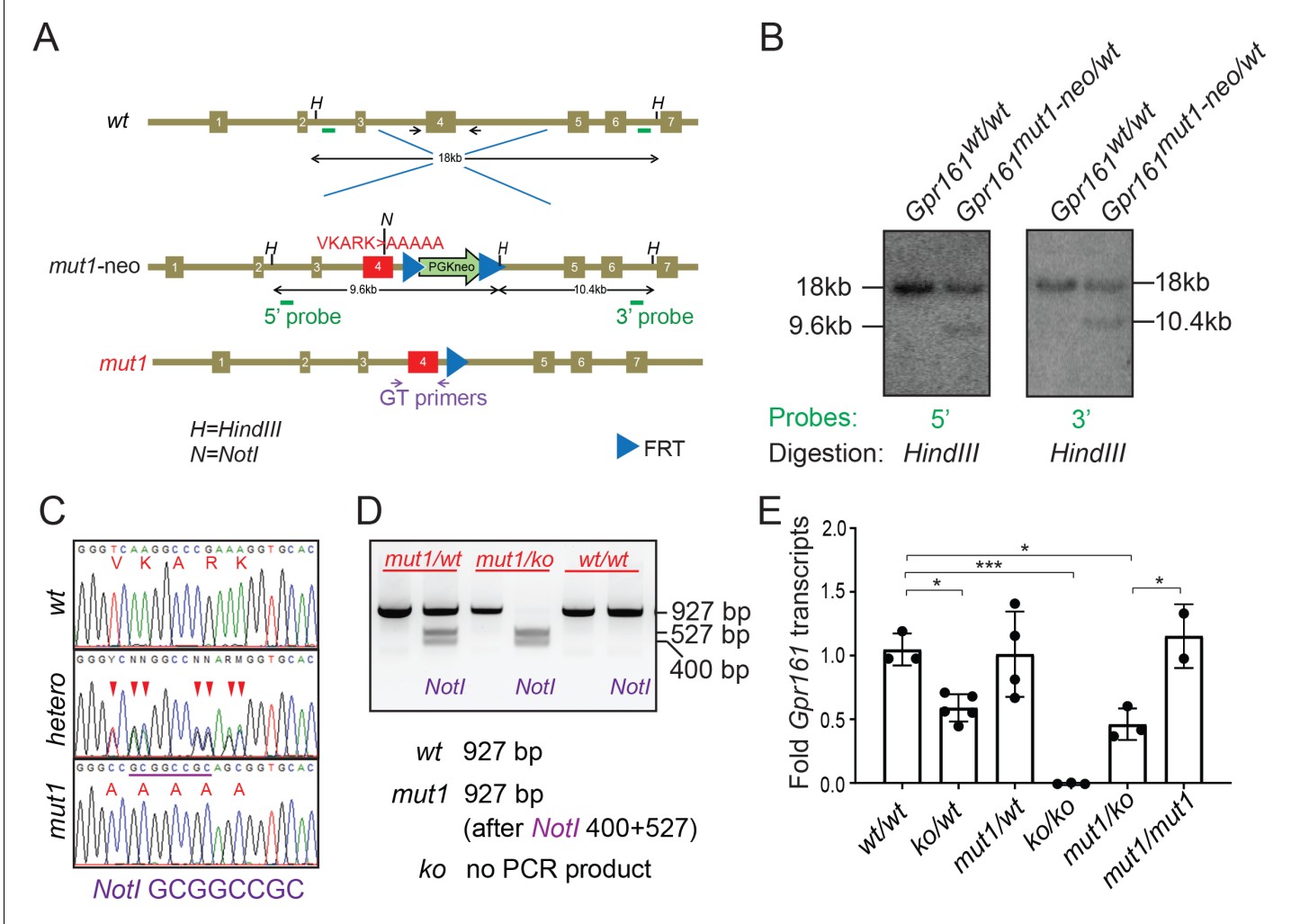

**Figure 2.** Generating ciliary localization defective endogenous knock-in *Gpr161mut1* mouse model. (**A**) The gene targeting strategy used to engineer the *Gpr161mut1* allele. Exons are numbered based on NM_001081126.2. PGKneo and FRT cassettes and genotyping (GT) primer sequences are indicated. The *mut1* sequence is located on Exon 4. (**B**) Southern blot analysis of representative ES cell clones using the 5' and 3' probes in A. (**C**) Sanger sequencing of *Gpr161wt* and *Gpr161mut1* alleles in adult mouse-tail DNA. Double peaks in *Gpr161wt/mut1* heterozygote indicated by arrowheads. The engineered NotI restriction site (GCGGCCGC) is indicated by a purple bar. (**D**) Genotyping for wild type, *Gpr161mut1* and knockout (ko) alleles by PCR using designated primers shown in A and digesting with NotI. (**E**) qRT-PCR of *Gpr161* transcripts normalized to *Hprt* in whole embryo extracts at E9.5 indicate diminished mRNA expression in the *Gpr161* knockout (*ko/ko*) embryos compared to wild type (*wt/wt*), but unchanged in *mut1/mut1* embryos. Data shown as mean ± SD. n=3 (*wt/wt*), 5 (*ko/wt*), 4 (*mut1/wt*), 3 (*ko/ko*), 3 (*ko/mut1*), 2 (*mut1/mut1*) embryos. *, $p<0.05$; ***, $p<0.001$. Other pairwise comparisons are not significantly different.

The online version of this article includes the following source data and figure supplement(s) for figure 2:

**Source data 1.** Table showing transcript data plotted in *Figure 2E*.

**Figure supplement 1.** Genomic DNA sequence and scheme of *Gpr161mut1* allele.

**Figure supplement 2.** Tandem affinity purification of LAP-tagged Gpr161wt or Gpr161mut1 (see Materials and methods) stably expressed in wild type MEFs was followed by immunoblotting for S-tag.

**Figure supplement 3.** The ciliary morphologies (**A**), including ciliary lengths (**B**) in *Gpr161ko/mut1* embryos at ~E10.25 in mesenchyme (MES), somatopleuric mesoderm (SP) or in the neural tube (NT) were unaffected compared to control wild type (*wt*) embryos.

microphthalmia (*Figure 3D–E*) and polydactyly in both forelimbs and hindlimbs (*Figure 3D*). Thus, the *Gpr161mut1* allele is hypomorphic to *Gpr161* ko. The *Gpr161mut1/mut1*, *Gpr161mut1/ko*, and *Gpr161ko/ko* embryos give rise to an allelic series increasing in phenotypic severity (*Table 2*).

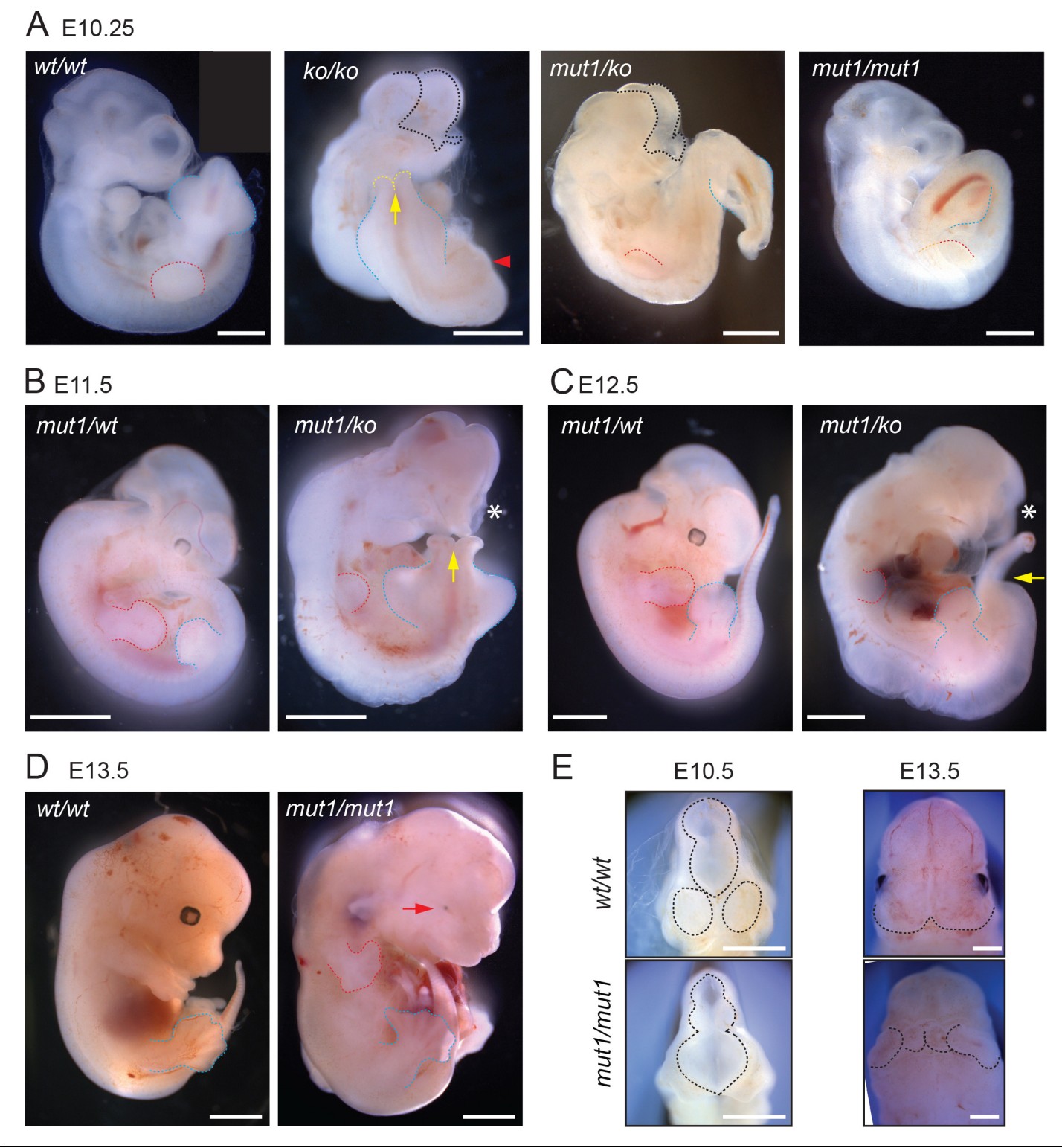

**Figure 3.** *Gpr161^{mut1}* allele is hypomorphic to knockout. (A – F) Bright-field images of wild type (*wt/wt*), *Gpr161* knockout (*ko/ko*), *mut1/ko* heterozygote, and *mut1* homozygote (*mut1/mut1*) at indicated time points. Red arrowhead indicates no limb bud in knockout embryo. *En face* view of E10.5 and E13.5 embryos in (E). Black dotted line or asterisk, rostral malformation; yellow arrow and dotted line, spina bifida; red dotted line, forelimb; blue dotted line, hindlimb. See also *Tables 1* and *2*. Scale: (A), 1 mm; (B–D), 2 mm; (D); (E), 1 mm.

**Table 1.** Results of breeding animals having *Gpr161 ko* and/or *mut1* alleles.

**Breeding between *Gpr161 ko/+* parents**

| Age | Litters | wt/wt | ko/wt | ko/ko |
|---|---|---|---|---|
| E8.5 | 16 | 36 (29%) | 54 (44%) | 33 (27%) |
| E9.5 | 26 | 51 (27%) | 85 (45%) | 54 (28%) |
| E10.25 | 8 | 16 (28%) | 27 (47%) | 15 (26%) |
| E10.5 | 6 | 14 (32%) | 30 (68%) | 0 (0%) ** |

**Breeding *Gpr161 ko/+* with *Gpr161 mut1/+* parents**

| Age | Litters | wt/wt | ko/wt or mut1/wt | mut1/ko |
|---|---|---|---|---|
| E9.5 | 12 | 24 (26%) | 45 (49%) | 22 (24%) |
| E10.25 | 9 | 20 (31%) | 28 (44%) | 16 (25%) |
| E10.5 | 2 | 3 (21%) | 7 (50%) | 4 (29%) |
| E11.5 | 2 | 3 (21%) | 7 (50%) | 4 (29%) |
| E12.5 | 5 | 14 (35%) | 23 (58%) | 3 (7%)[¥] |
| E13.5 | 4 | 8 (36%) | 14 (64%) | 0 (0%)[†, *] |

**Breeding between *Gpr161 mut1/+* parents**

| Age | Litters | wt/wt | mut1/wt | mut1/mut1 |
|---|---|---|---|---|
| E9.5 | 20 | 41 (30%) | 61 (45%) | 34 (25%) |
| E10.25 | 4 | 8 (25%) | 16 (50%) | 8 (25%) |
| E10.5 | 8 | 21 (30%) | 34 (49%) | 15 (21%) |
| E11.5 | 2 | 6 (43%) | 4 (29%) | 4 (29%) |
| E12.5 | 7 | 16 (29%) | 27 (49%) | 12 (22%) |
| E13.5 | 9 | 16 (27%) | 29 (48%) | 15 (25%) |
| E14.5 | 7 | 13 (30%) | 23 (52%) | 8 (18%)[‡] |
| E14.75 | 6 | 13 (35%) | 24 (65%) | 0 (0%)[§, **] |
| E15.5 | 5 | 10 (38%) | 16 (62%) | 0 (0%)[¶, ***] |

[¥]3 dead *mut1/ko* embryos were not counted.

[†]7 dead *mut1/ko* embryos were not counted.

[‡]7 dead *mut1/mut1* embryos were not counted.

[§] 13 dead *mut1/mut1*embryos were not counted.

[¶] Dead or resorbed embryos were not counted.

* p<0.05 by Chi Square test.

** p<0.01 by Chi Square test.

*** p<0.001 by Chi Square test.

## *Gpr161*[mut1/ko] embryos have reduced Hh pathway hyperactivation compared to knockouts

As *Gpr161*[mut1] allele is later embryonic lethal compared to knockout, we tested if Hh signaling is impacted in *Gpr161* mutants by analyzing the expression of direct transcriptional targets of the Hh pathway, *Gli1* and *Patched2* (*Ptch2*). In whole embryo extracts at E9.5, we detected increased levels of both transcripts in the *Gpr161*[ko/ko] embryos, intermediate levels in *Gpr161*[mut1/ko] embryos, and slightly elevated levels but statistically insignificant from wild type in *Gpr161*[mut1/mut1] embryos (*Figure 4A–B*). Gli1 protein levels also showed similar trends (*Figure 4C*). These data indicate that *Gpr161* knock-in mutants experience gradually decreasing levels of Hh pathway hyperactivity compared to ko, commensurate with monoallelic or biallelic expression of *Gpr161*[mut1].

## *Gpr161*[mut1/ko] embryos have intermediate GliR levels compared to knockouts

In the absence of Hh ligand, PKA-mediated phosphorylation of Gli3 results in its limited proteolysis into Gli3R (*Niewiadomski et al., 2014*; *Tempé et al., 2006*; *Wang et al., 2000*). In *Gpr161*[ko/ko]

**Table 2.** Phenotypes in *Gpr161* mutants.

| Phenotype (age) | *Gpr161*<sup>ko/ko</sup> | *Gpr161*<sup>ko/mut1</sup> | *Gpr161*<sup>mut1/mut1</sup> |
|---|---|---|---|
| Lethality of embryos | E10.5 | E13.5 | E14.75 |
| Exencephaly (>E10.25) | 100% (102/102) | 100% (54/54) | 100% (12/12) |
| Spina bifida (>E10.25) | 100% (15/15) | 100% (16/16) | N/D |
| Midface widening (>E12.5) | N/A | N/A | 100% (23/23) |
| Microphthalmia (>E12.5) | N/A | 100% (3/3) | 100% (35/35) |
| Pericardial effusion (E12.5–13.5) | N/A | 92% (11/12) | N/D |
| Limb development | | | |
| Forelimb | No forelimbs at E10.25 | Stunted at E12.5 | Polydactyly at E13.5 (6~7 digits) |
| Hindlimb | Present | Present | Polydactyly at E13.5 (6~7 digits) |
| Kinked tail (> E12.5) | N/A | 100% (3/3) | N/D |

N/A, not applicable; N/D, not determined.

embryos, the extent of Gli3 processing into Gli3R at E9.5 whole-embryonic extracts was strongly decreased compared to wild type, suggesting that Gpr161 directs processing of Gli3 into Gli3R (*Hwang et al., 2018*; *Mukhopadhyay et al., 2013*). Compared to wild-type, the extent of Gli3 processing into Gli3R at E9.5 whole-embryonic extracts was strongly decreased in *Gpr161*<sup>ko/ko</sup> and *Gpr161*<sup>ko/mut1</sup> embryos, and trended toward decreased levels, but were not statistically significant in the *Gpr161*<sup>mut1/mut1</sup> embryos. Upon Hh signaling limited proteolysis of Gli3 is prevented, whereas activated full-length Gli proteins are formed that are unstable (*Chen et al., 2009*; *Humke et al., 2010*; *Jia et al., 2009*; *Wen et al., 2010*; *Wang et al., 2010*). Compared to wild type, Gli3 full-length forms trended toward decrease in *Gpr161*<sup>ko/mut1</sup> and *Gpr161*<sup>mut1/mut1</sup> embryos but were not statistically significant (*Figure 4C*). Gli3 full-length to Gli3R ratios were significantly increased in both *Gpr161*<sup>ko/ko</sup> and *Gpr161*<sup>ko/mut1</sup> embryos with respect to wild type (*Figure 4C*). Gli2R levels were only moderately reduced in *Gpr161*<sup>ko/ko</sup> embryos at E9.5, while being less affected in the *Gpr161*<sup>ko/mut1</sup> and *Gpr161*<sup>mut1/mut1</sup> embryos (*Figure 4—figure supplement 1*). In summary, the *Gpr161* mutants exhibit gradually decreasing severity of Gli3 processing defects compared to ko, commensurate with monoallelic or biallelic expression of *Gpr161*<sup>mut1</sup>.

## *Gpr161*<sup>mut1</sup> NIH 3T3 cells exhibit Gli2 accumulation in cilia similar to *Gpr161* knockouts

Upon Hh pathway activation, Gli2 proteins accumulate at the tips of cilia (*Chen et al., 2009*; *Haycraft et al., 2005*; *Kim et al., 2009*). While ~20% of cilia tips in control cells had detectable Gli2 staining, treatment with SAG increased Gli2 accumulation in control cilia in NIH 3T3 cells (*Figure 4D–E*). In contrast, ~60% of *Gpr161* ko cilia in NIH 3T3 cells had Gli2 staining irrespective of SAG treatment (*Figure 4D–E*). Stable expression of wild type *Gpr161* (*Gpr161*<sup>wt</sup>), but not that of *Gpr161*<sup>mut1</sup>, rescued the basal Gli2 ciliary accumulation in *Gpr161* ko cells. Thus, basal Gli2 accumulation in ciliary tips occurs from the absence of ciliary Gpr161 pools.

## *Gpr161*<sup>mut1/ko</sup> embryos exhibit less ventralized neural tube compared to *Gpr161* knockouts

The neural tube is patterned during early embryonic development by Hh secreted from the notochord. The neuroprogenitors acquire different spatial identities along the dorso-ventral axis due to a complex interplay of transcription factors, which are expressed in response to relative variations in Hh levels, as well as the duration for which they are exposed (*Dessaud et al., 2008*). As we had demonstrated before (*Mukhopadhyay et al., 2013*), Hh-dependent ventral cell types were ectopically specified at the expense of lateral and dorsal cell types (ventralization) in the *Gpr161*<sup>ko/ko</sup> at E9.5 (*Figure 5*). Specifically, floor plate progenitors expressing FoxA2, and p3/pMN/p2 progenitors expressing Nkx6.1 showed enlarged expression domains, expanding into comparatively more dorsal

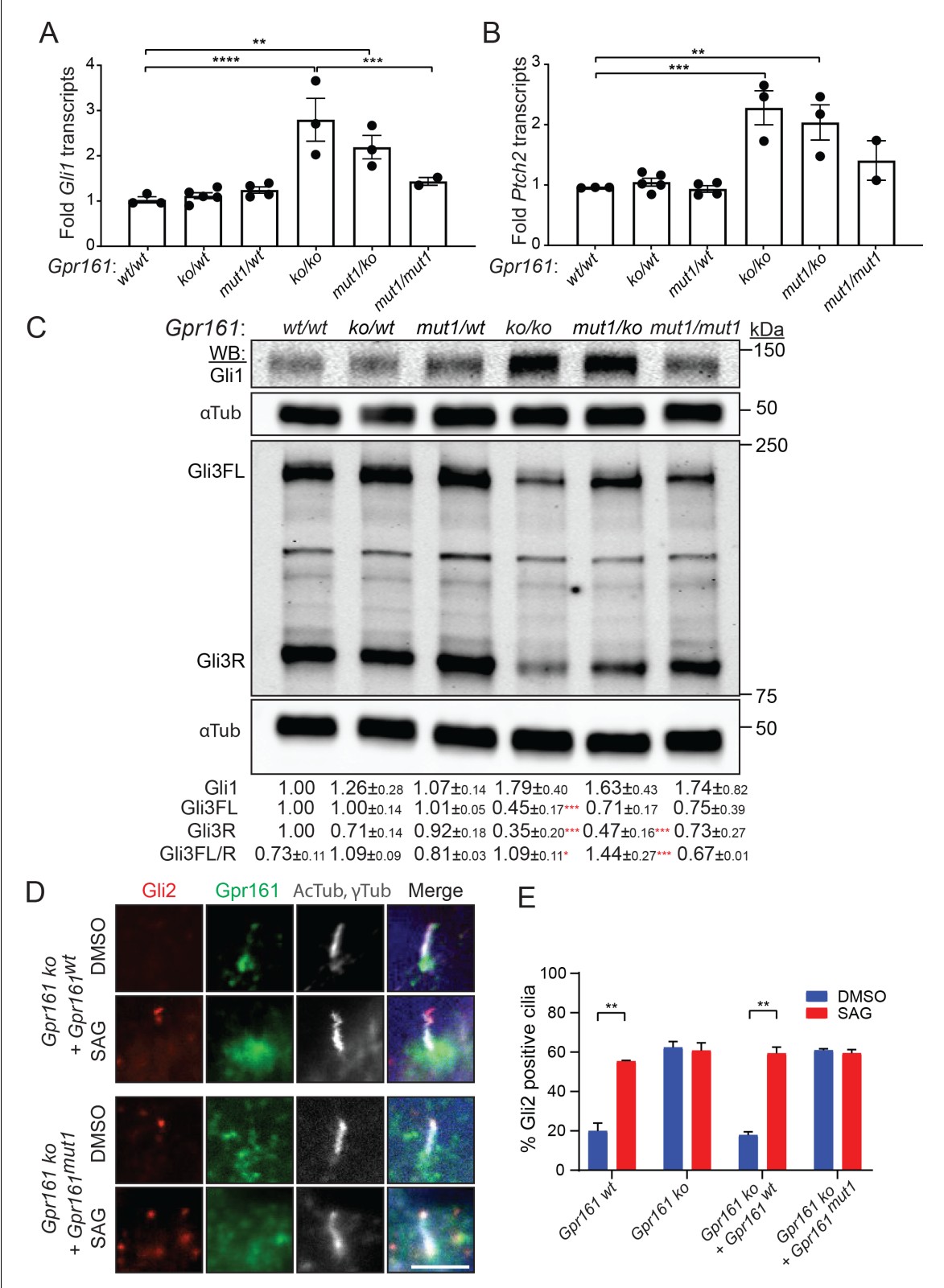

**Figure 4.** *Gpr161^mut1/ko* embryos have reduced Hh pathway hyperactivation compared to knockouts. (**A – B**) *Gli1* (**A**) and *Ptch2* (**B**) transcript levels in whole-embryo extracts at E9.5 by qRT-PCR, normalized to *Gapdh*, n = 3–4 embryos each, data shown as mean ± SEM. **, p<0.01, ***, p<0.001, ****, p<0.0001. Only significant differences are marked. (**C**) Immunoblotting for Gli1, Gli3 and α-tubulin in whole-embryo lysates at E9.5. n=2 or 3 independent experiments for Gli1 or Gli3 immunoblotting, respectively. Data shown as mean ± SD normalized to α-tubulin. *, p<0.05; **, p<0.01; ***,

*Figure 4 continued on next page*

*Figure 4 continued*

p<0.001; ****, p<0.0001. (**D**) NIH 3T3 Flp-In *Gpr161* ko cells stably expressing untagged wildtype (*wt*) or *Gpr161^mut1^* were starved for 24 hr upon confluence and were treated for further 24 hr ± SAG (500 nM). After fixation, cells were immunostained with anti-Gli2 (red), anti-Gpr161 (green), anti-acetylated, and γ-tubulin (AcTub, γTub in grey) antibodies. (**E**) Quantification of Gli2 positive cilia from (**D**). n=100 cells counted/condition from two coverslips each. Data shown as mean ± SD. **, p<0.01. Scale: (**E**), 5 µm.

The online version of this article includes the following source data and figure supplement(s) for figure 4:

**Source data 1.** Table showing transcript data plotted in *Figure 4A and B*.
**Source data 2.** Table showing data from individual experiments in *Figure 4C*.
**Source data 3.** Table showing Gli2 positive cilia data plotted in *Figure 4E*.
**Figure supplement 1.** Immunoblotting for Gli2 and α-tubulin in E9.5 whole-embryo lysates.
**Figure supplement 1—source data 1.** Table showing data from individual experiments i *Figure 4—figure supplement 1*.

regions throughout the rostrocaudal extent of the spinal cord and hindbrain (*Figure 5*, *Figure 5—figure supplement 1*). Pax6 was expressed in dorsally restricted domains, whereas Pax7 was strongly downregulated (*Figure 5*).

Ventral-most neural tube patterning is dependent on the downstream activation of Gli2 (*Bai and Joyner, 2001*), whereas intermediate-level patterning is Gli3R regulated (*Persson et al., 2002*). We investigated if the *Gpr161* ko phenotype is dependent on Gli2 by generating *Gpr161; Gli2* double ko. We generated linked double mutants by recombination, as *Gpr161* and *Gli2* genes are on the same chromosome. *Gli2* single ko is perinatal lethal (*Bai and Joyner, 2001*). The *Gpr161; Gli2* double ko survived till E12.75, past the embryonic lethality period for *Gpr161* single ko at E10.5. The double ko embryos also exhibited exencephaly, similar to *Gpr161* ko (*Figure 5*), suggesting a possible role for Gli3 in this region (*Liu et al., 2015*; *Shimada et al., 2019*; *Yu et al., 2009*). *Gli2* ko shows sparse FoxA2 expression in the floorplate throughout the rostrocaudal extent of the neural tube without affecting Nkx6.1, compared to wild type (*Somatilaka et al., 2020*). The double ko had reduced FoxA2 in floor plate compared to *Gpr161* ko, suggesting that ventralization requires Gli2A. However, persistent but limited ventralization of Nkx6.1 domain suggested independence from Gli2A (*Figure 5*, *Figure 5—figure supplement 1*). The dorsolateral markers Pax7 and Pax6 were both partially restored compared to *Gpr161* ko (*Figure 5*). Another negative regulator that we recently described, Ankmy2, an ankyrin repeat and MYND domain protein, functions by regulating adenylyl cyclase trafficking to cilia (*Somatilaka et al., 2020*). Ventral expansion of the floor plate progenitors in *Ankmy2* knockout, even though being more severe than *Gpr161* knockout, is fully Gli2-dependent. However, expansion of Nkx6.1 in the intermediate region of the neural tube persisted upon concomitant Gli2 loss in *Ankmy2* knockout, suggesting a role of lack of Gli3R. Thus, ventralization of FoxA2 in *Gpr161* ko is Gli2-dependent, whereas Nkx6.1 domain ventralization is likely to be Gli3R-dependent.

As we had detected a partial hyperactivation of Hh targets and decrease in Gli3R levels in *Gpr161^ko/mut1^* whole embryo lysates (*Figure 4*), we next compared the neural tube phenotypes of *Gpr161^ko/mut1^* and *Gpr161^mut1/mut1^* embryos with *Gpr161^ko/ko^* and *Gpr161; Gli2* double ko. In contrast to *Gpr161^ko/ko^*, *Gpr161^ko/mut1^* embryos had normal FoxA2 expression limited to floor plate. However, Nkx6.1 was expanded dorsally, although lesser in extent than in *Gpr161^ko/ko^*. Pax6 was not dorsally restricted, whereas Pax7 was restored, unlike *Gpr161^ko/ko^* (*Figure 5*). Neural tube patterning in *Gpr161^mut1/mut1^* embryos was almost similar to wild type, except for a more restricted mid-dorsal expression of Pax7 (*Figure 5*). Thus, ventral-most FoxA2 patterning was unaffected in *Gpr161^ko/mut1^* and *Gpr161^mut1/mut1^* embryos, unlike *Gpr161* ko, suggesting that ventral-most progenitor expansion that requires Gli2A does not occur upon loss of ciliary Gpr161 pools. The persistence of limited Nkx6.1 ventralization in *Gpr161^ko/mut1^* and *Gpr161; Gli2* double ko phenocopies lack of Gli3 repressor (*Persson et al., 2002*). The Nkx6.1 domains were unaffected in *Gpr161^mut1/mut1^* embryos (*Figure 5*, *Figure 5—figure supplement 1*) from likely restoration of Gli3R at this embryonic stage (*Figure 4C*). Thus, ciliary pools of Gpr161 are required for generating sufficient Gli3R levels to prevent such intermediate level ventralization. Increased extraciliary Gpr161 pools in *Gpr161^mut1/mut1^* also prevent Nkx6.1 ventralization from likely restoration of Gli3R processing.

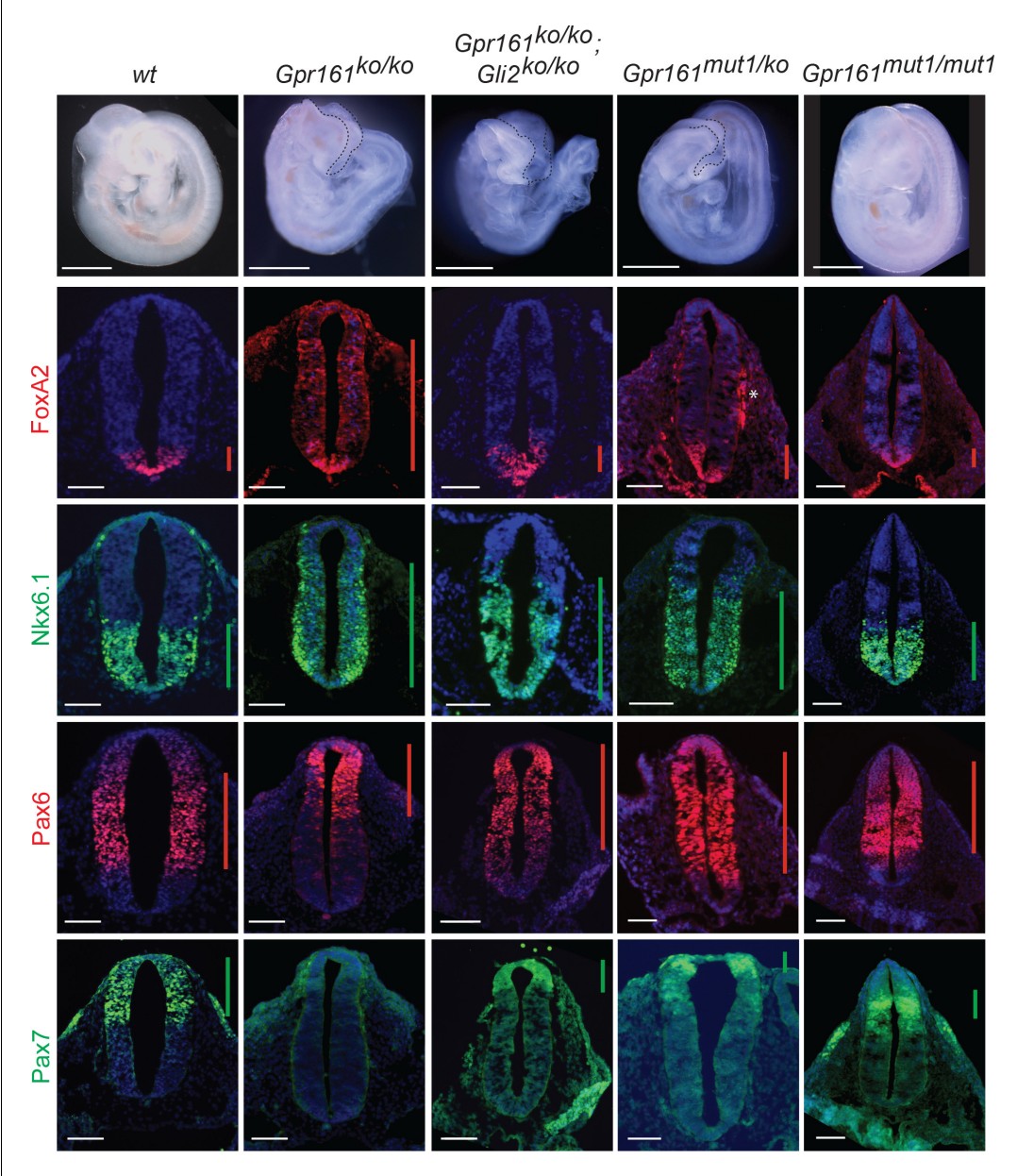

**Figure 5.** *Gpr161^{mut1/ko}* embryos exhibit less ventralized neural tube compared to *Gpr161* knockouts. Topmost panels show bright-field images of wildtype (*wt*), *Gpr161 ko*, *Gpr161 mut1/ko*, *Gpr161 mut1/mut1*, *Gpr161; Gli2* double *ko*, and *Gli2 ko* whole-mount embryos at E9.5. Bottom panels show rostral neural tube horizontal sections immunostained using designated markers. All images are counterstained with Hoechst. Black dotted line mark rostral malformations. Vertical bars show the extent of dorsoventral expression of markers. Asterix, nonspecific background staining outside neural tube. n=2–4 embryos each genotype and immunostaining. Scale: 50 μm.

The online version of this article includes the following figure supplement(s) for figure 5:

**Figure supplement 1.** Quantification of dorsoventral (D–V) extent of Nkx6.1 expression *in Gpr161* mutants.

## *Gpr161^{mut1/mut1}* embryos exhibit high Hh signaling and polydactyly in limb buds

As neural tube patterning defects were minimal in *Gpr161^{mut1/mut1}* embryos, we tested if other tissues affected by Hh signaling were affected. *Shh* is expressed from the zone of polarizing activity (ZPA) in posterior limb buds from E9.75 (*Charité et al., 2000*) and patterns the digits (*Niswander, 2003*). Mutual antagonism between Gli3R and the bHLH transcription factor, dHand,

likely prepatterns the mesenchyme before *Shh* expression (*te Welscher et al., 2002*). dHand also promotes Shh expression from ZPA (*Charité et al., 2000*).

*Gpr161$^{ko/ko}$* embryos lack forelimbs and have increased Shh signaling in the hind limbs (*Hwang et al., 2018*). Depletion of *Gpr161* in the limb mesenchyme using *Prx1-Cre* (*Prx1-Cre; Gpr161$^{f/f}$*) results increased expression of *Ptch1/Gli1* in both anterior and posterior forelimb fields and ectopic expression of Shh anteriorly. In contrast to *Gpr161$^{ko/ko}$*, forelimb buds were formed in *Gpr161$^{mut1/mut1}$* embryos, and as in *Prx-Cre; Gpr161$^{f/f}$* there was increased expression of *Ptch1/Gli1* in both anterior and posterior forelimb and hindlimb fields, as detected by RNA in situ hybridization (ISH) (*Figure 6A*). Several features suggested a lack of Gli3R antagonism in causing Shh pathway hyperactivation. First, Shh was diffusely expressed posteriorly at E10.5 as detected by RNA ISH (*Figure 6B*), a likely result of dHand misexpression from Gli3R lack. Ectopic anterior expression of Shh in *Prx1-Cre; Gpr161$^{f/f}$* is also likely from Gli3R lack that manifests later in the limb mesenchyme (*Hwang et al., 2018*). Second, *Hoxd13* expression by RNA ISH was anteriorly expanded in *Gpr161$^{mut1/mut}$* embryos at E10.5 (*Figure 6C*), a likely result of lack of Gli3R antagonism (*te Welscher et al., 2002*; *Niswander, 2003*). Third, we noticed a reduction of Gli3R formation in limb buds of *Gpr161$^{mut1/mut1}$* embryos by immunoblotting at E12.5 (*Figure 6D*).

By E13.5, *Prx1-Cre; Gpr161$^{f/f}$* had increased number of digit fields resulting in polysyndactylous phenotypes (*Hwang et al., 2018*; *Figure 6E*). *Gpr161$^{mut1/mut1}$* embryos also showed polydactyly in both forelimb and hindlimb buds as apparent from RNA ISH for *Sox9* and *Col2a1* for mesenchymal and chondrogenic condensations, respectively (*Figure 6E*). Thus, lack of Gli3R in *Gpr161$^{mut1/mut1}$* embryos is sufficient to cause increased Shh signaling and polydactyly.

## *Gpr161$^{mut1/mut1}$* embryos show high Hh signaling and mid face widening

Similar to limb bud patterning we tested if other tissues affected by lack of Gli repressor are similarly affected from loss of ciliary pools of Gpr161. Loss of cilia in cranial neural crest cells causes midfacial widening, duplicated nasal septum and bilateral cleft secondary palate (*Brugmann et al., 2010b*; *Chang et al., 2016*). Deletion of both *Gli2* and *Gli3* phenocopies these phenotypes, which are restored by expression of Gli3R (*Chang et al., 2016*), suggesting that ciliary signaling regulate GliR formation that is important in morphogenesis of this tissue. While lack of cilia in *Wnt1-Cre; Kif3a$^{f/f}$* mutants shows lack of Gli2/3R levels, expression of *Ptch1* and *Gli1* are asynchronized, and are not reflective of the lack of repression (*Chang et al., 2016*).

Cranial neural crest-specific deletion of *Gpr161* using *Wnt1-Cre* resulted in microphthalmia, severe midfacial cleft and widening (*Figure 7A,D*). Similar midfacial widening as evident from increased distance between nasal pits was also observed in the *Gpr161$^{mut1/mut1}$* embryos (*Figure 7B–E*). The midfacial widening resulted from increased gap between maxillary processes by ingression of median nasal processes (*Figure 7B–C*). *Gpr161$^{mut1/mut1}$* embryos had high Hh signaling in craniofacial structures including nasopharangeal processes, palatal shelves and tongue, as evident from high *Ptch1* transcript levels by RNA ISH (*Figure 7B*). We also noted increased *Gli1* levels in the tongue of *Gpr161$^{mut1/mut1}$* embryos by RNA ISH (*Figure 7C*). *Gpr161; Gli2* double ko embryos also showed midfacial widening and high *Ptch1* transcript levels by RNA ISH in the nasopharangeal processes (*Figure 7F*). Thus, high Hh signaling and midfacial widening from loss of Gpr161 ciliary pools was not Gli2A-dependent but was likely from GliR lack.

## Discussion

### *Gpr161$^{mut1}$* establishes an allelic series in regulating Hh pathway derepression

Hh signaling is regulated by downstream GliA and GliR levels. GliA/R are generated from the same parental template, which makes their formation inter-dependent. Thus, GliA formation is inherently dependent on suppressed GliR processing (*Pan et al., 2006*) and phenotypes from GliR processing defects, such as in *Gpr161/Ankmy2* ko or in *PKA* null embryos can result from GliR loss or GliA formation (*Kopinke et al., 2021*). We generated a Gpr161$^{mut1}$ receptor variant that did not transit through cilia but demonstrated constitutive cAMP signaling activity like wild type. The gradually decreasing severity of Hh pathway transcriptional upregulation and Gli3R processing defects among

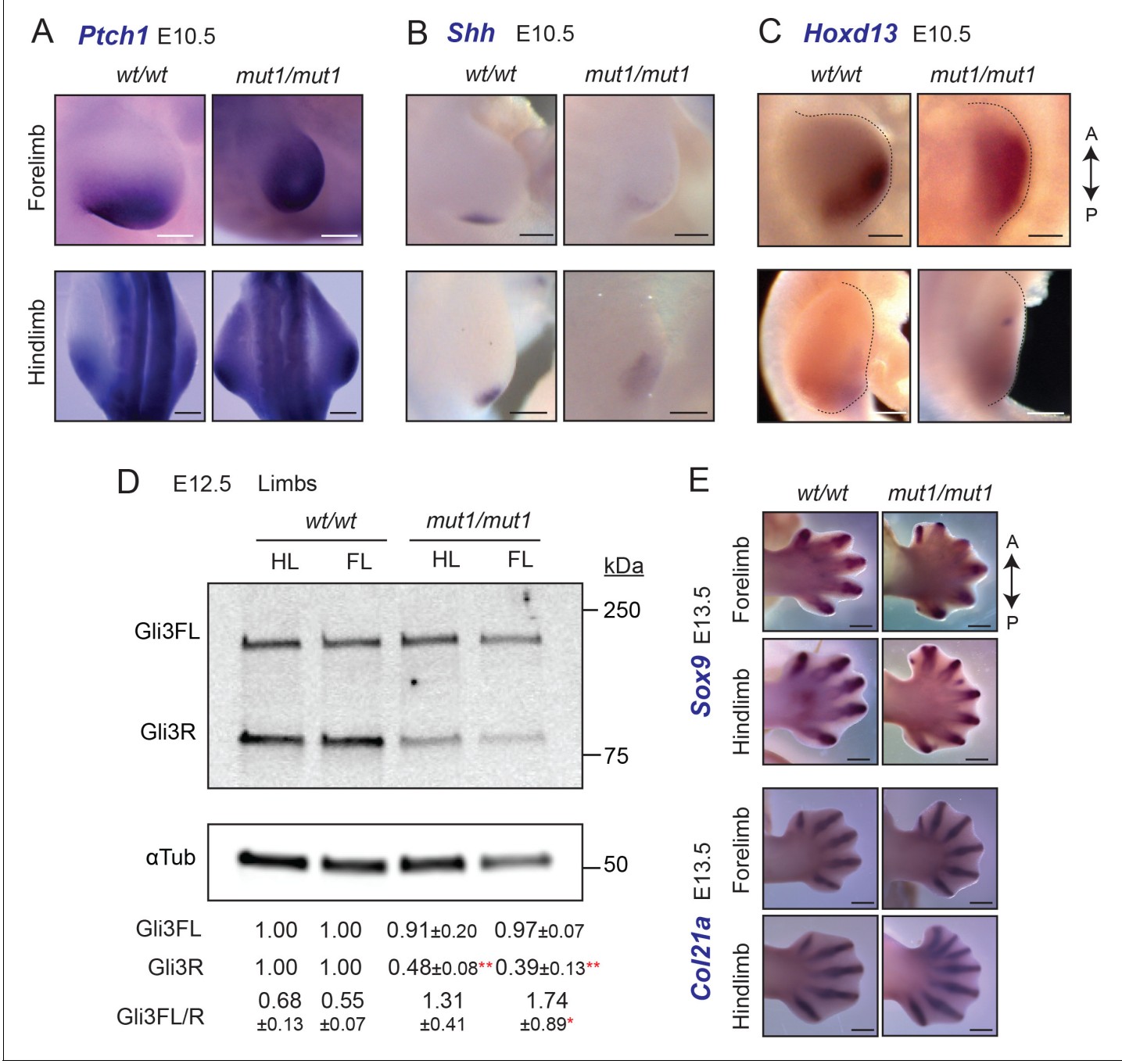

**Figure 6.** *Gpr161^{mut1/mut1}* embryos exhibit high Hh signaling and polydactyly in limb buds. (A–C) RNA in situ hybridization for E10.5 *Ptch1* (A), *Shh* (B) *Hoxd13* (C) in limb buds. *Ptch1* and *Hoxd13* were expanded anteriorly in *Gpr161^{mut1/mut1}* limb buds. Shh was diffusely expressed posteriorly but was not ectopically expressed anteriorly in *Gpr161^{mut1/mut1}* limb buds. n=2–5 each. (D) Immunoblotting of forelimb (FL) and hindlimb (HL) buds for Gli3 and α-tubulin shows decreased Gli3R levels at E12.5 wildtype (wt) versus *Gpr161^{mut1/mut1}*. Quantification shown is normalized to α-tubulin. n = three experiments. *, p<0.05; **, p<0.01. (E) RNA in situ hybridization for *Sox9* and *Col2a1* in E13.5 wildtype and *Gpr161^{mut1/mut1}* limb buds. *Gpr161^{mut1/mut1}* limb buds show polydactyly. n=four each. Scale: (A–C), 50 µm; (E), 500 µm.

The online version of this article includes the following source data for figure 6:

**Source data 1.** Table showing data from individual experiments in *Figure 6D*.

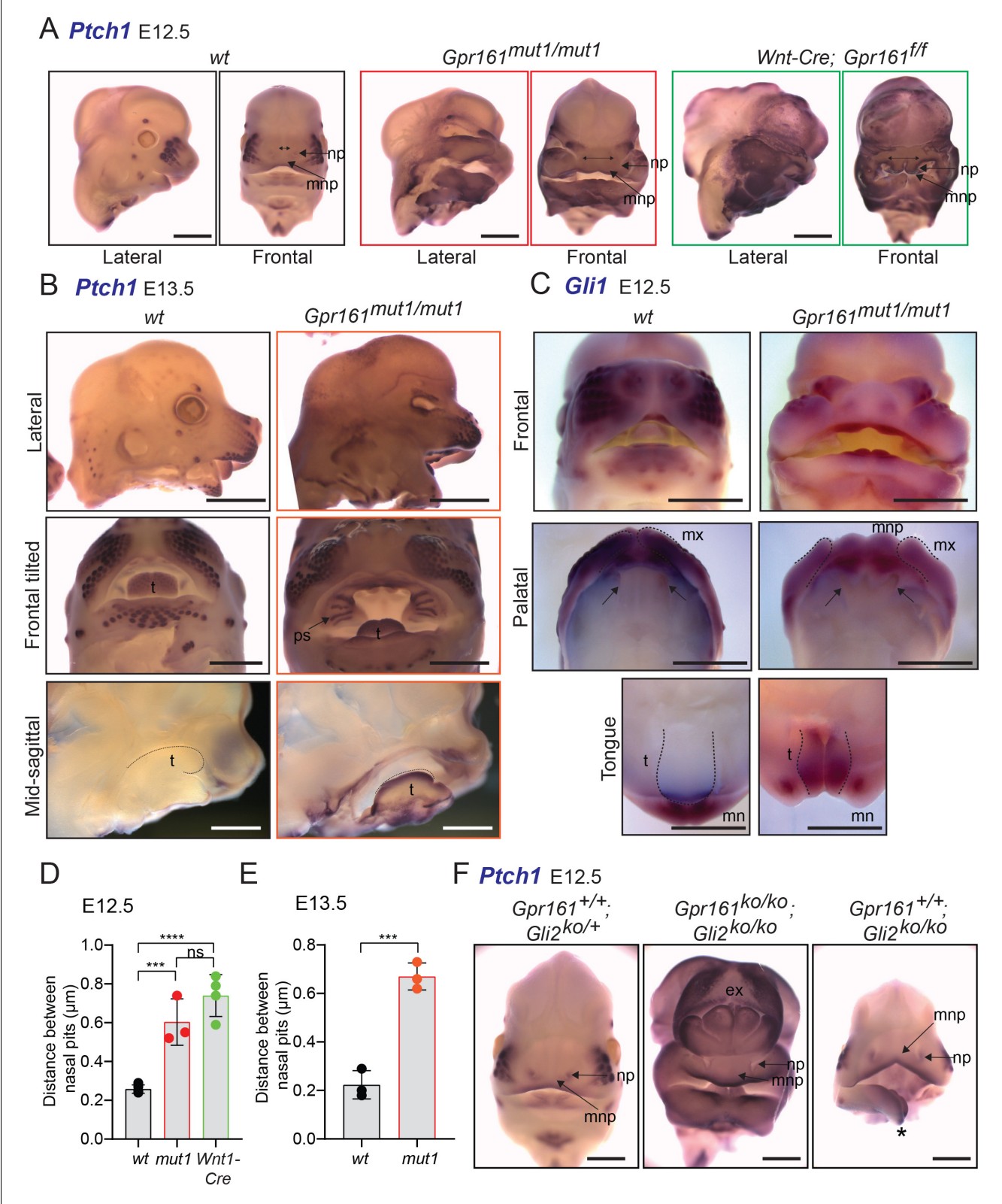

**Figure 7.** *Gpr161^mut1/mut1* embryos show high Hh signaling and mid face widening. (**A**) RNA in situ hybridization for *Ptch1* in wildtype (*wt*), *Gpr161^mut1/ mut1*, and *Wnt1-Cre; Gpr161^f/f* E12.5 head. Distances between nasal pits are shown as bidirectional arrows. Lateral and frontal views are shown. n=3–4 each. (**B**) RNA in situ hybridization for *Ptch1* in E13.5 wildtype (*wt*), *Gpr161^mut1/mut1* head. Top panels, lateral view; middle panels, frontal tilted view shows palate and tongue; bottom panels, sagittal section showing tongue. Black dotted lines in the bottom panels indicate tongue. Bidirectional

*Figure 7 continued on next page*

Figure 7 continued

arrows show increased distance between nasal pits. Arrow points to prominent palatal shelf in *Gpr161*$^{mut1/mut1}$. n=3 each. (**C**) RNA in situ hybridization for *Gli1* in E12.5 wildtype (*wt*), *Gpr161*$^{mut1/mut1}$ head. Top panels, frontal view. Middle panels show palates imaged from below (palatal view) and bottom panels show lower jaw viewed from above (tongue view) after separating the jaws. Arrows in upper panel show secondary palatal shelves. Black dotted lines in the bottom panels indicate tongue. Note increased gap between maxillary processes by ingression of median nasal processes. n=3 each. (**D–E**) Quantification of distance between nasal pits as shown in (**A**). The colors are matched with each strain in A and B. Error bars represent SEM. ***, p<0.001; ****, p<0.0001, unpaired t-test. n=3–4 each. (**F**) RNA in situ hybridization for *Ptch1* in E12.5 control (*Gli2*$^{-/+}$), *Gpr161; Gli2* double ko and *Gli2* ko head. Note persistent exencephaly in *Gpr161; Gli2* double ko and midfacial widening. Displaced lower jaw is an artifact (*). n=1–2 each. Scale: (**A** and **F**), 1 mm; (**B** and **C**) 2 mm. Abbreviations: ex, exencephaly; mnp, medial nasal process; mx, maxillary process; mn, mandibular process; np, nasal pit; ps, palatal shelf; t, tongue.

The online version of this article includes the following source data for figure 7:

**Source data 1.** Table showing data for distance between nasal pits plotted in *Figure 7D and E*.

*Gpr161* ko, *Gpr161*$^{mut1/ko}$ and *Gpr161*$^{mut1/mut1}$ embryos establishes an allelic series for *Gpr161*, showing that both ciliary and extraciliary receptor pools function as a rheostat in regulating Hh signaling strength. The distinct phenotypic consequences in different tissues, however, argues for threshold specific response to GliA/R levels. Ventral-most progenitor expansion in neural tube is seen in absence of Gpr161 and is Gli2-dependent. Lack of Gpr161 ciliary pools specifically was incapable of generating sufficient GliA required for ventral-most progenitor expansion (*Figure 8A*). Nkx6.1 ventralization in the intermediate neural tube occurred from loss of ciliary Gpr161 (*Figure 8A*), but such expansion was prevented by increased extraciliary Gpr161$^{mut1}$ levels that generated Gli3R. Tissue regions that depend predominantly on GliR were also specifically affected from loss of ciliary pools of Gpr161. These regions included the developing limb buds and face (*Figure 8B–C*). Collectively, effects of ciliary loss of Gpr161 pools are tissue specific and are likely dependent on the requirements of the tissues on GliR vs GliA in morphogenesis. Tissue-specific Gli3R thresholds established by ciliary and extraciliary Gpr161 likely dictate morpho-phenotypic outcomes.

## Limitations of the *Gpr161*$^{mut1}$ allele

Currently, a limitation of the *Gpr161*$^{mut1}$ allele arises from the fact that a physiological ligand, if any, for Gpr161 is unknown. Thus, we are unable to test role of Gpr161$^{mut1}$ in ligand-induced cAMP signaling and formally rule out if this mutant is functioning as a hypomorph, in addition to being extraciliary. There have been conflicting reports of ciliary cAMP concentration in kidney epithelial cells ranging from levels comparable to that in the cytoplasm (*Jiang et al., 2019*) to supraphysiological levels ~4.5 µM (*Moore et al., 2016*). In the absence of a known ligand for Gpr161, and lack of consensus regarding efficacy of the currently available ciliary targeted cAMP probes, we are unable to show impact of Gpr161$^{mut1}$ on ciliary cAMP levels.

Another limitation of any approach that perturbs subcellular pools of cAMP, be it through genetic, optogenetic or chemogenetic means (*Guo et al., 2019*; *Hansen et al., 2020*; *Truong et al., 2021*), arises from the cilioplasm not being isolated from the cytoplasm. The second messengers, cAMP and Ca$^{2+}$, are both freely diffusible between ciliary and extraciliary compartments (*Delling et al., 2016*; *Truong et al., 2021*). Therefore, extraciliary cAMP could diffuse into ciliary compartments and vice versa. The contribution of the leaked cAMP from extraciliary cAMP production in *Gpr161*$^{mut1/mut}$ in the consequent phenotypes is unknown. However, the concentrations achieved from such leaks might not reach critical thresholds required for downstream PKA signaling at cilia (*Truong et al., 2021*).

A recent study used elegant optogenetic methods regulating subcellular cAMP levels and genetic approaches subcellularly expressing dominant negative PKA to perturb Shh-mediated somite patterning in zebrafish (*Truong et al., 2021*). They showed that ciliary, but not cytoplasmic, production of cAMP functions through PKA localized in cilia to repress Shh-mediated somite patterning. We show that lack of ciliary Gpr161 pools result in Hh hyperactivation phenotypes in specific tissues or regions that are specifically impacted by GliR. However, we also find that the ventral neural tube is unaffected in patterning from lack of ciliary Gpr161 pools, presumably from deficient Gli2A formation that requires complete Gpr161 loss. Overall, our results are strikingly complementary to the optogenetic study by showing that lack of ciliary Gpr161 pools result in Hh hyperactivation

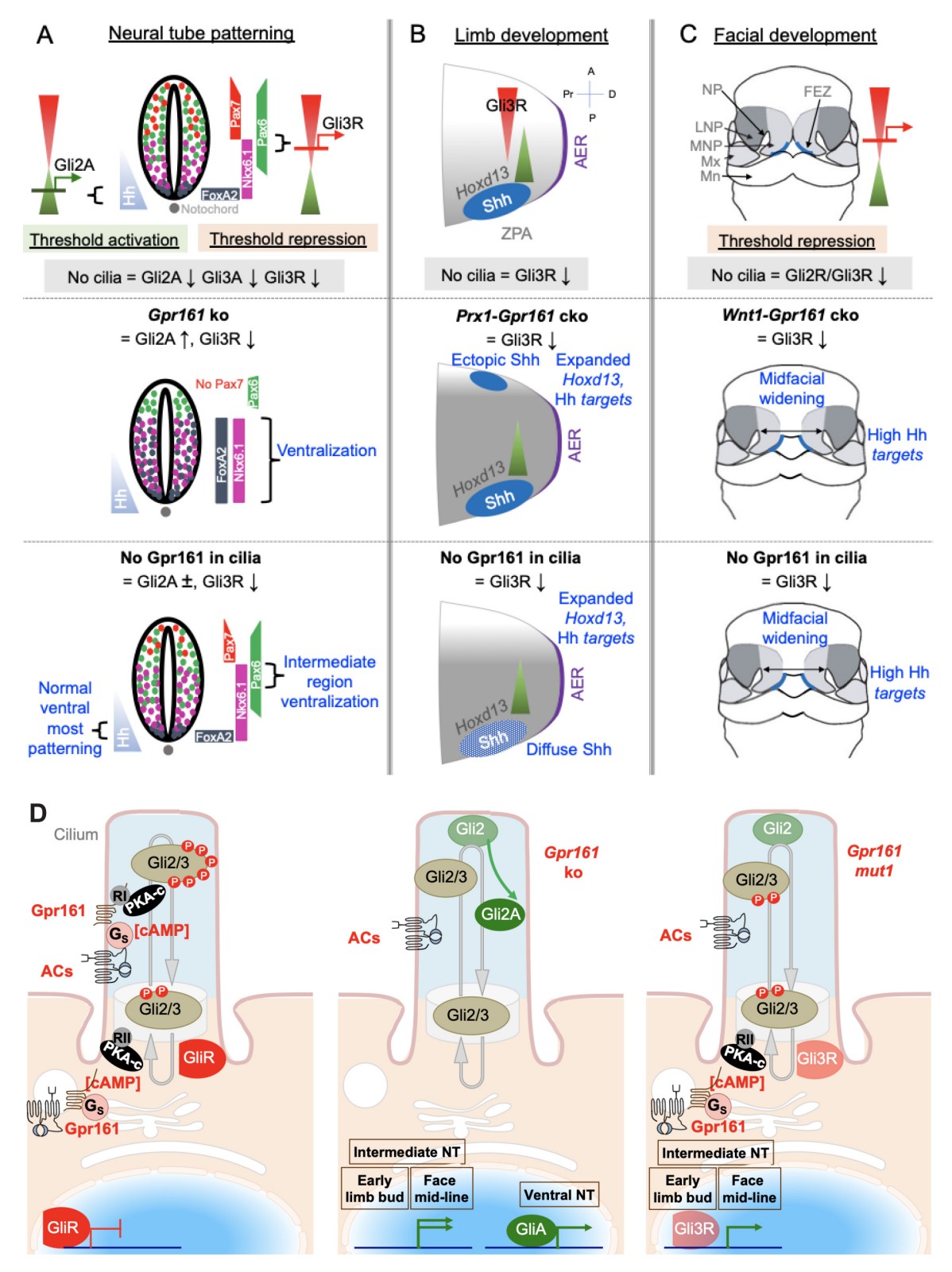

**Figure 8.** Gpr161 ciliary pools determine Hh pathway repression-regulated morpho-phenotypic spectrum. (**A**) Neural tube development. Hh is expressed from the notochord. Gli2A-mediated threshold activation mediates floorplate and ventral-most progenitor patterning. Gli3R regulates intermediate level patterning. Ciliary disruption prevents patterning of all ventral progenitors. Complete loss of Gpr161 in ko causes ventralization of all ventral progenitors from excessive Gli2A generation and loss of Gli3R. Lack of Gpr161 ciliary pools in *Gpr161*$^{mut1/ko}$ reduces Gli3R formation sufficiently

*Figure 8 continued on next page*

Figure 8 continued

to cause intermediate level expansion of Nkx6.1 but does not generate excessive GliA to cause ventralization of floor plate markers. Caudal spina bifida and exencephaly persists in *Gpr161^{mut1/ko}* similar to *Gpr161* ko. (B) Limb development. Shh expression from ZPA starting from E9.75 establishes posterior gradient of pathway targets such as *Ptch1/Gli1*. Anterior Gli3R gradient also limits expression of genes such as *Hoxd13* to the posterior mesenchyme. Ciliary disruption causes preaxial polydactyly from lack of Gli3R induced anterior expansion of *5'Hoxd* gene expression. *Gpr161* ko prevents forelimb formation. Conditional deletion of *Gpr161* in limb mesenchyme using *Prx1-Cre; Gpr161^{f/f}* (*Prx1-Gpr161* cko) causes increased Hh pathway targets, reduced Gli3R, and expanded *Hoxd13* in limb buds contributing to polydactyly. Loss of Gpr161 ciliary pools in *Gpr161^{mut1/mut1}* causes increased Hh pathway targets, reduced Gli3R, and expanded *Hoxd13* in limb buds contributing to polydactyly. Shh expression in the posterior limb bud is diffuse, likely from lack of counter-antagonism between Gli3R and dHand. Abbreviations: AER, anterior ectodermal ridge; ZPA, zone of polarizing activity; A, anterior; P, posterior; Pr, proximal; D, distal. (C) Facial development. Shh is expressed from the frontonasal ectodermal zone in the medial nasal processes. Threshold repression by both Gli2R and Gli3R prevents midfacial widening. Ciliary disruption, or lack of both Gli2/3 causes midfacial widening, which is prevented by forced Gli3R expression. Loss of Gpr161 ciliary pools in *Gpr161^{mut1/mut1}* phenocopies *Gpr161* deletion in craniofacial mesenchyme using *Wnt1-Cre; Gpr161^{f/f}* (*Wnt1-Gpr161* cko) by showing midfacial widening and increased levels of Shh pathway targets. Gli2 loss is unable to rescue midfacial widening and increased Shh pathway targets in *Gpr161* ko background, suggesting reduced GliR contributing to these phenotypes. Abbreviations: FEZ, frontonasal ectodermal zone; LNP, lateral nasal process; MNP, medial nasal process, Mx, maxillary process. (D) Model of Gpr161 ciliary and extraciliary pools in Hh pathway. Both ciliary and extraciliary pools of Gpr161 contribute to GliR formation by PKA-mediated phosphorylation. Complete loss of Gpr161 prevents Gli2/3 phosphorylation by PKA. Gli2 is less efficient in repressor formation than Gli3. Unphosphorylated Gli2 is dissociated from Sufu and accumulates in ciliary tips. GliA formation is dependent on lack of PKA phosphorylation but likely occurs downstream of Gli2 ciliary accumulation. Smo might further promote GliA formation. Gli2 accumulation in ciliary tips and GliA formation also occurs in *Ankmy2* knockout where trafficking of ACs to cilia is affected. Loss of ciliary Gpr161 restricts ciliary cAMP-mediated PKA activation but does not hinder extraciliary Gpr161 from cAMP production. Such extraciliary production might happen in the endomembrane compartment, but PKA regulatory subunits involved are not known. Restricted phosphorylation limits Gli3R formation but is still sufficient to dissociate Gli2 from Sufu causing accumulation in ciliary tips. However, GliA formation is impacted. Tissue regions that depend on GliR are specifically affected from loss of ciliary pools of Gpr161. Abbreviations: RI/RII, type I/II PKA regulatory subunits.

phenotypes arising likely from lack of GliR in the limb buds, mid-face and intermediate neural tube. Currently, we have not demonstrated conclusively that the resultant phenotypes in *Gpr161^{mut1}* are from loss of Gli3 repressor. The commonly used *Gli3R* allele (*Gli3Δ699*) expresses a fusion protein containing the first 699 aa of Gli3 and extra residues from thymidine kinase used for generating the allele (*Böse et al., 2002*). The *Gli3Δ699* allele is of limited potency as a Gli3R, because it does not resemble *Shh* ko during limb development (*Cao et al., 2013*) as would be expected from Gli3R antagonizing Shh activity (*Litingtung et al., 2002*; *Persson et al., 2002*). Introducing improved *Gli3R* alleles with potency equal to that of the natural repressor (*Gli3Δ701*) (*Cao et al., 2013*) in the background of *Gpr161* and/or *Gli2* mutants will be required to conclusively show role of GliR in *Gpr161^{ko}/Gpr161^{mut1}* phenotypes.

Subcellular expression of Gpr161 variants is, however, likely to lead to increased specificity of localized cAMP action compared to optogenetic and chemogenetic approaches increasing diffusible cAMP. The textbook model of cAMP causing dissociation of PKA catalytic subunits from regulatory subunits (*Walsh and Cooper, 1979*) is unlikely to occur in subcellular contexts based on the following evidence. First, at physiological concentrations of cAMP (1–2 µM), most PKA catalytic subunits remain associated with regulatory subunits (*Smith et al., 2017*). Full dissociation occurs only at supraphysiological concentration. Second, physiological agonists causing native production of cAMP does not promote catalytic subunit release (*Smith et al., 2017*). Third, PKA signaling is sustained when catalytic and regulatory subunits are constrained (*Smith et al., 2017*). Fourth, released catalytic subunits are also rapidly recaptured by regulatory subunits after activation (*Mo et al., 2017*; *Walker-Gray et al., 2017*). Instead, the PKA regulatory subunit-AKAP complexes are fundamentally important in organizing and sustaining PKA catalytic subunit activation for localized substrate phosphorylation in restrictive nanodomains (*Bock et al., 2020*; *Zhang et al., 2020*). The dual functions of Gpr161 in Gαs coupling and as an atypical AKAP is likely to further restrict cAMP-PKA signaling in microdomains.

## Gpr161 ciliary and extraciliary pools cumulatively contribute to Gli-repressor formation

Gpr161 pools both inside and outside cilia cumulatively contribute to Gli3R processing, with higher extraciliary pools in *Gpr161^{mut1/mut1}* embryos increasing processing compared to *Gpr161^{mut1/ko}* embryos at E9.5 (*Figure 8D*). Despite contribution from both ciliary and extraciliary pools of Gpr161

in Gli3R processing, neural tube ventralization in *Gpr161* ko is completely dependent on cilia. Therefore, Gpr161 activity outside cilia in the in vivo settings is probably through the periciliary recycling endosomal compartment where it localizes in addition to cilia (*Mukhopadhyay et al., 2013*) and could activate ACs and PKA in proximity to the centrosome (*Figure 8D*). Production of cAMP by Gpr161 in the endomembrane compartment, as reported for other GPCRs (*Calebiro et al., 2009*; *Crilly and Puthenveedu, 2021*; *Irannejad et al., 2013*; *Kotowski et al., 2011*; *Vilardaga et al., 2014*), could be relevant in cumulatively contributing to Gli3R processing. Direct Gpr161 C-tail binding to type I PKA regulatory subunits (*Bachmann et al., 2016*) is likely to facilitate PKA activation in proximity to Gpr161-mediated cAMP production in cilia for GliR processing (*May et al., 2021*; *Mick et al., 2015*), but if such coupling happens in the endomembrane compartment is not known.

*Gpr161^{mut1/mut1}* limb buds showed decreased Gli3R processing compared to whole embryo lysates, suggesting limited compensation by extracellular Gpr161 pools in the limb buds and tissue specificity in contribution of extracellular Gpr161 pools to Gli3R processing. Phosphorylation by PKA in six Gli2/3 C-terminal clusters further primes phosphorylation by glycogen synthase kinase 3β and Casein kinase 1 (*Tempé et al., 2006*). Currently, subcellular location for these sequential phosphorylation events in Gli3R processing is not known (*Kopinke et al., 2021*). Precise cataloging of the phosphorylation cluster modifications in individual tissues would be required to determine tissue specificity in Gli3R processing efficiency.

## Gpr161 subcellular pools uncouple Gli2 accumulation in cilia from activator formation

C-term phosphorylation of Gli2/3 proteins causes GliR formation, whereas GliA formation is dependent on lack of C-term phosphorylation by PKA (*Niewiadomski et al., 2014*). Expressing a *Gli2* mutant that is PKA-nonphosphorylatable has been shown to induce chick neural tube ventralization (*Niewiadomski et al., 2014*). Gli2A formation is compromised in *Gpr161^{mut1/mut1}* embryos in the neural tube. While Gli2R formation is much less efficient than Gli3R formation, lack of Gpr161 both inside and outside cilia is likely required for preventing Gli2 phosphorylation by PKA and resultant Gli2A formation. However, *Gpr161* ko cells show Gli2 accumulation in cilia, and such accumulation persists upon *Gpr161^{mut1}* expression in ko cells upstream of full scale Gli2 activation (*Figure 8D*). Gli2/3 is complexed with Suppressor of Fused (Sufu) and GliR formation requires Gli-Sufu complex (*Humke et al., 2010*; *Kise et al., 2009*). Dissociation of Sufu from Gli2/3 is a hallmark of Hh-stimulated signaling and requires cilia (*Humke et al., 2010*; *Tukachinsky et al., 2010*). Gli2 accumulation in cilia probably arises from dissociation from Sufu of the limited phosphorylated Gli2, which still undergoes processing. Similar basal accumulation of Gli2 in cilia has also been observed in resting *PKA* null MEFs (*Tuson et al., 2011*) and cells lacking Ankmy2 that regulates adenylyl cyclase trafficking to cilia (*Somatilaka et al., 2020*).

## Gpr161 ciliary pools regulate Hh repression mediated morpho-phenotypic spectrum

Conceptually, the complexity in Hh regulation in tissues by cilia can be simplified based on genetic epistasis between Hh pathway and *Gli2*, *Gli3R* and cilia mutants (*Falkenstein and Vokes, 2014*; *Kopinke et al., 2021*). Phenotypes rescued by Gli3R or Gli2 loss are GliR- or Gli2A- dependent, exemplifying tissues regulated primarily by GliA or GliR thresholds, respectively. Phenotypes rescued by both Gli3R or Gli2 loss exemplify tissues that sense ratio between GliA and GliR levels, whereas repression by GliR in absence of Hh expression is a feature of other tissues. Tissues that predominantly require Gli2/Gli3 activation for patterning, such as the ventral neural tube, show lack of activation from loss of cilia. In contrast, tissues that require Gli repressor, such as midface and limbbuds, show lack of repression phenotypes with loss of cilia. Although Gli family members bind to same consensus DNA sequences, the differential roles of GliA and GliR in sculpting tissues could likely arise from context-dependent cis-regulation of targets by co-activators/repressors (*Oosterveen et al., 2012*). Alternatively, tissue-specific regulation of enhancer activity could regulate Gli-mediated repression (*Lex et al., 2020*). The effects of ciliary loss of Gpr161 are dependent on requirements of the tissues on GliR levels as follows:

## Neural tube

Ventral part of the neural tube is mainly regulated by Gli2 and Gli3 activators (*Bai and Joyner, 2001*; *Bai et al., 2004*), whereas the intermediate region of the neural tube is regulated by Gli3R levels (*Persson et al., 2002*). Complete lack of *Gpr161* causes expansion of the ventral, including ventral-most progenitors rostrocaudally (*Mukhopadhyay et al., 2013*). However, lack of Gpr161 ciliary pools in *Gpr161^{mut1/ko}* prevent ventral-most progenitor expansion but retains ventral expansion of Nkx6.1 in the intermediate neural tube. Thus, lack of ciliary Gpr161 pools in *Gpr161^{mut1/ko}* prevent sufficient Gli3R processing required to inhibit Nkx6.1 ventralization (*Figure 8A*) and phenocopies *Gpr161; Gli2* double ko. Furthermore, restoration of Gli3R by increased extracellular Gpr161 pools in *Gpr161^{mut1/mut1}* prevents ventralization of Nkx6.1.

*Gpr161^{mut1/ko}* embryos also show exencephaly and spina bifida, like *Gpr161* ko (*Mukhopadhyay et al., 2013*). Recently, *GPR161* mutations have been reported in patients suffering from caudal neural tube defects (spina bifida) (*Kim et al., 2019*). While mutants causing Hh pathway hyperactivation are associated with neural tube defects (*Murdoch and Copp, 2010*; *Somatilaka et al., 2020*; *Ybot-Gonzalez et al., 2002*), cross-regulation with other pathways including Wnt (*Zhao et al., 2014*) and BMP signaling (*Ybot-Gonzalez et al., 2007*) could determine neural tube closure at different spinal levels. A ciliary retained but PKA-RI non-binding *Gpr161^{vl}* truncation mutant (*Bachmann et al., 2016*; *Pal et al., 2016*) is also associated with spina bifida (*Matteson et al., 2008*) and could involve Wnt signaling defects (*Li et al., 2015*). Role of ciliary Gpr161 in regulation of these pathways with relation to neural tube closure is presently unclear.

## Limb buds

Lack of Gpr161 ciliary pools in limb buds increased Hh pathway targets and expanded *Hoxd13* from lack of Gli3R, which likely contributes to the polydactyly. We do not see ectopic Shh expression as in *Prx1-Cre; Gpr161^{f/f}* (*Hwang et al., 2018*). Instead, Shh transcripts in the posterior limb bud is diffuse, probably arising from deficient counter-antagonism between Gli3R and dHand—that drives Shh expression (*Charité et al., 2000*; *te Welscher et al., 2002*). Similarly, a hypomorphic *Ift88* mutant—that causes short cilia causes expansion of *dHand*, suggesting compromised Gli3R formation. This mutant manifests preaxial polydactyly without ectopic Shh expression (*Liu et al., 2005*). Forelimb buds are also completely lacking in *Gpr161^{ko/ko}* embryos (*Hwang et al., 2018*), unlike in *Gpr161^{mut1/ko}* and *Gpr161^{mut1/mut1}* embryos, likely from high Hh signaling in the presumptive limb fields. Therefore, lack of Gpr161 in cilia demonstrate effects from moderate levels of derepression of the Hh pathway (*Figure 8B*).

## Craniofacial development

Shh is expressed from the frontonasal ectodermal zone (FEZ) in the medial nasal processes (*Hu and Marcucio, 2009*). Midfacial tissues are regulated by Gli2R/Gli3R levels in normal morphogenesis by preventing midfacial widening (*Chang et al., 2016*; *Schock and Brugmann, 2017*). Lack of Gpr161 ciliary pools phenocopies *Gpr161* deletion in cranial neural crest using *Wnt1-Cre* in showing midfacial widening. Loss of cilia or lack of Gli2 and Gli3 also shows midfacial widening (*Brugmann et al., 2010a*; *Chang et al., 2016*). The facial phenotypes are accompanied by increased levels of *Ptch1* and *Gli1* transcripts in the mid-facial tissues. However, *Ptch1* and *Gli1* expression are not increased upon ciliary loss (*Chang et al., 2016*), suggesting Hh hyperactivation to be stronger upon *Gpr161* loss from cilia. Cilia regulates both GliA and GliR; therefore, loss of cilia, despite causing lack of repression, might not be as effective in causing pathway hyperactivation as *Gpr161* loss. The craniofacial phenotypes and upregulation of *Ptch1* transcripts persists in mid face of *Gpr161; Gli2* double ko, consistent with derepression from lack of GliR in causing the mid-facial phenotypes (*Figure 8C*).

In conclusion, the results presented here suggest a role for Gpr161-mediated signaling propagated by ciliary and extraciliary compartments in establishing tissue specific GliR thresholds regulating morpho-phenotypic outcomes from Hh pathway hyperactivation. By establishing an allelic series for Gpr161-mediated Hh pathway repression, we are also poised to unravel neomorphic phenotypic outcomes from varying signaling strength of Hh signaling.

## Materials and methods

### Experimental model and subject details

#### ES cells

A BAC overlapping the mouse *Gpr161* genomic locus was engineered by recombineering using an FRT-PGKneo-FRT selection cassette. The required mutation (with an engineered NotI site) was separated from the selection cassette by 536 bp (*Lee et al., 2001*). Recombineering was done by the Biomedical Research Core Facility, University of Michigan (*Zeidler et al., 2011*). The exon 4 fragments of the *Gpr161^mut1* targeting construct were generated using this engineered BAC (#RP23-235-E18) by further cloning into a modified pGKneoloxP2.DTA.2 vector as follows. The pGKneo-loxP2.DTA.2 vector was digested with SacII and NheI and religated. The left arm consisting of 4100 bp upstream and the right arm consisting of 4800 bp downstream of the FRT-PGKneo-FRT cassette in the BAC were PCR cloned into the HindIII site of this modified vector (*Figure 2—figure supplement 1*). The engineered plasmid was linearized with XhoI and was electroporated into JM8.N4 ES cells at the Transgenic Core in UT Southwestern, Dallas (*Figure 2C*). From among 469 ES cell clones, 56 clones were first screened by PCR, and two clones 4D9 and 5F9 were further screened by southern blotting with probes as marked in *Figure 2A*. ES cells were grown on SNL feeders with media containing 20% Serum, 6 mM L-glutamine, 1X Penicillin/Streptomycin, 1 mM β-mercaptoethanol, 1 mM Non-essential Amino Acids, 1X Nucleosides, 10 mg/L Sodium Pyruvate, ESGRO supplement 66 µl/L and incubated at 37˚C in 5% $CO_2$ (Dr. Robert Hammer lab, UT Southwestern, Dallas).

### Mouse strains

The *Gpr161^{mut1-neo/+}* ES cells (5F9 clone) were injected into host embryos of the C57BL/6 albino strain by the transgenic core (Dr. Robert Hammer lab, UT Southwestern Medical Center, Dallas). Mice with germline transmission were crossed with Flp-O (Jackson lab; Stock no: 012930) for deleting the FRT-PGKneo-FRT cassette to generate the *Gpr161^mut1* allele. The *Gpr161* knockout and conditional allele targeting the third exon crossed has been described before (*Hwang et al., 2018*). Double knockout analysis was performed using *Gli2^{tm1Alj}* (ko) allele (*Mo et al., 1997*). *Prx1-Cre* (*Logan et al., 2002*; Jax strain No: 005584) or *Wnt1-Cre* (*Lewis et al., 2013*; Jax strain No: 022501) was crossed with the *Gpr161^{f/f}*. The *Gli2 ko* and *Gpr161* floxed alleles were linked through genetic recombination by breeding *Gpr161^{f/f}* with *Gli2^{ko/+}* animals. Crossing with CAG-Cre recombinase line (*Sakai and Miyazaki, 1997*), in which Cre is expressed ubiquitously, generated the linked *Gpr161; Gli2* double knockout allele. Yolk sac DNA was used for genotyping embryos. Mice were housed in standard cages that contained three to five mice per cage, with water and standard diet ad libitum and a 12 hr light/dark cycle. Noon of the day on which a vaginal plug was found was considered E0.5. All the animals in the study were handled according to protocols approved by the UT Southwestern Institutional Animal Care and Use Committee, and the mouse colonies were maintained in a barrier facility at UT Southwestern, in agreement with the State of Texas legal and ethical standards of animal care.

### Cell lines and MEFs

NIH3T3 and RPE hTERT were authenticated by and purchased from ATCC. They have tested negative for Mycoplasma. β-arrestin 1/2 double knockout MEFs were from Robert Lefkowitz and tested negative for Mycoplasma. The *Gpr161* ko NIH 3T3 Flp-In cell line was a gift from Rajat Rohatgi (*Pusapati et al., 2018*). The cells were cultured in DMEM-high glucose media (D5796; Sigma) with 10% BCS (Sigma-Aldrich), 0.05 mg/ml penicillin, 0.05 mg/ml streptomycin, and 4.5 mM glutamine. Stable knockout cell lines were generated by retroviral infection with pBABE constructs having untagged wild type or mutant *Gpr161* inserts followed by antibiotic selection. RPE hTERT cells were grown in DMEM F12 media with 10% FBS (Sigma-Aldrich), 0.05 mg/ml penicillin, 0.05 mg/ml streptomycin, and 4.5 mM glutamine. RPE hTert cells were retrovirally infected with C-term LAP-tagged *Gpr161* wild type or mutant constructs and flow sorted for GFP after puromycin selection. Stable doxycycline-inducible NIH 3T3 Tet-on 3 G cells expressing untagged *Gpr161* and *Gpr161^mut1* receptor variants were generated by retroviral infection. These cell lines were grown in Tet-free serum (Clontech) until induction by Doxycycline (*Mukhopadhyay et al., 2013*). Single, multiple amino acid

mutations, or deletions in full-length Gpr161 were generated using Quikchange site-directed muta-genesis kit (Stratagene or Q5 Mutagenesis Kit (NEB)).

## Method details

### Mouse genotyping

Genotyping of *Gpr161 mut1* alleles were performed using primers in the intron 3–4 (5′ CAGAAAG-CAACAGCAAAGCA) and intron 4–5 (5′ ACCCTGACACTGCCCTTAGC). The PCR product of wild type and *mut1* allele bands was 927 bp, but only the PCR product from the *mut1* allele was digested into 400 and 527 bp products with NotI. Genotyping of *Gpr161* knockout or floxed alleles were per-formed using primers in the deleted 4th exon (5′ CAAGATGGATTCGCAGTAGCTTGG), flanking the 3′ end of the deleted exon (5′ ATGGGGTACACCATTGGATACAGG), and in the Neo cassette (5′ CAACGGGTTCTTCTGTTAGTCC). Wild type, floxed and knockout bands were 816, 965, and 485 bp, respectively (*Hwang et al., 2018*). *Cre* allele was genotyped with Cre-F (5′-AAT GCT GTC ACT TGG TCG TGG C-3′) and Cre-R (5′-GAA AAT GCT TCT GTC CGT TTG C-3′) primers (100 bp ampli-con). To genotype *Gli2* mice, Gli2 sense (5′-AAA CAA AGC TCC TGT ACA CG-3′), Gli2 antisense (5′-CAC CCC AAA GCA TGT GTT TT-3′) and pPNT (5′-ATG CCT GCT CTT TAC TGA AG-3′) primers were used. Wild type and knockout bands were 300 bp and 600 bp, respectively.

### siRNA transfection

RPE hTERT cells were passaged on glass coverslips and transfected with siRNA using Lipofectamine RNAiMax (Invitrogen). OTP non-targeting pool (GE Healthcare) was used as control siRNAs in all experiments. The siRNA sequences for INPP5E (J-020852–05) is 5′-GGAAUUAAAAGACGGAUUU-3′ and has been validated before (*Humbert et al., 2012*). The siRNA sequences for BBS4 (D-013649–04) is 5′-CGAUCUGACUUAUAUAAUG-3′ and has been validated before (*Loktev et al., 2008*). For INPP5E RNAi, 100 nM siRNA was transfected during plating followed by 100 nM reverse transfection 24 hr after plating. Seventy-two hr after first transfection, the cells were starved with 0.2% FBS starv-ing media for 24 hr and fixed in 4% paraformaldehyde (PFA) for immunofluorescence. For *BBS4* RNAi, 100 nM siRNA was reverse transfected 24 hr after plating. 48 hr after the transfection, the cells were starved with 0.2% FBS starving media for 24 hr and fixed in 4% paraformaldehyde (PFA) for immunofluorescence.

### TR-FRET cAMP assays

We used a TR-FRET cAMP assay (Cisbio) for estimating cAMP as these assays eliminate the back-ground noise due to FRET-based estimation of signals. Also, the use of long-lived fluorophores (in this case lathanides), combined with time-resolved detection (a delay between excitation and emis-sion detection) minimizes prompt intrinsic fluorescence interferences (*Degorce et al., 2009*). TR-FRET assays on stable doxycycline-inducible NIH 3T3 Tet-on 3 G cells expressing *Gpr161* or *Gpr161mut1* were performed according to manufacturer's instructions (Cisbio) (*Mukhopadhyay et al., 2013*). Cells were plated on 96-well poly-D-lysine coated plates and induced with doxycycline (2 ug/ml) for 24 hr. Cells were finally treated with the nonselective phosphodiester-ase inhibitor IBMX (1 mM). Individual treatments were run in each experiment in triplicate. The final FRET counts were recorded using an EnVision 2103 multiplate reader (Perkin Elmer). cAMP levels (nM) were calculated as interpolated values from a standard curve. Standard curves were generated using least squares fitting method (with $R^2$ value >0.99) and values for unknowns were interpolated using Graphpad Prism.

### Tissue processing, antibodies, immunostaining, and microscopy

Mouse embryos fixed in 4% PFA overnight at 4°C and processed for cryosectioning. For cryosection-ing, the embryos were incubated in 30% sucrose at 4°C until they were submerged in the solution. Embryos were mounted with OCT compound. Embryos in OCT were cut into 15 μm frozen sections. The sections were incubated in PBS for 15 min to dissolve away the OCT. Sections were then blocked using blocking buffer (1% normal donkey serum [Jackson immunoResearch, West Grove, PA] in PBS) for 1 hr at room temperature. Sections were incubated with primary antibodies against the following antigens; overnight at 4°C: FoxA2 (1:1000, ab108422; Abcam), Nkx6.1 (1:100, F55A10-s; DSHB), Pax6 (1:2000, 901301; Biolegend), Pax7 (1:10; DSHB). After three PBS washes, the sections

were incubated in secondary antibodies (Alexa Fluor 488-, 555-, 594-, 647- conjugated secondary antibodies, 1:500; Life Technologies, Carlsbad, CA or Jackson ImmunoResearch) for 1 hr at room temperature. Cell nuclei were stained with Hoechst 33342 (Life Technologies). Slides were mounted with Fluoromount-G (0100–01; Southern Biotech) and images were acquired with a Zeiss AxioImager.Z1 microscope. For immunofluorescence experiments in cell lines, cells were cultured on coverslips until confluent and starved for 48 hr. To quantify ciliary Gli2 and Gpr161 levels, cells were treated with 500 nM SAG or DMSO for 24 hr after 24 hr of serum starvation. In some experiments, Transferrin conjugated with Alexa Fluor 568 (Life technologies, T23365) was added (10 µg/ml) in starvation media at 37°C for 30 min before fixation. Cells were fixed with 4% PFA for 10 min at room temperature and postfixed for 5 min with methanol at −20°C for γ-tubulin immunostaining. After blocking with 5% normal donktey serum, the cells were incubated with primary antibody solutions for 1 hr at room temperature followed by treatment with secondary antibodies for 30 min along with Hoechst. Primary antibodies used were against Gpr161 (1:200, custom-made) (*Pal et al., 2016*), acetylated α-tubulin (mAb 6-11B-1, Sigma; 1:2000), GFP (Abcam ab13970), Tulp3 (1:500, gift from Jonathan Eggenschwiler) (*Norman et al., 2009*), Gli2 (1:500, gift from Jonathan Eggenschwiler) (*Cho et al., 2008*), γ-tubulin (GTU-88, Sigma; 1:500). Coverslips were mounted with Fluoromount-G and images were acquired with a Zeiss AxioImager.Z1 microscope using a 40 × oil immersion objective lens.

## Reverse transcription, quantitative PCR

RNA was extracted using the GenElute mammalian total RNA purification kit (RTN350; Sigma). Genomic DNA was eliminated by DNase I (D5307; Sigma). qRT-PCR was performed with SYBR Green Quantitative RT-qPCR Kit (QR0100; Sigma) or Kicqstart One-Step Probe RT-qPCR ReadyMix (KCQS07; Sigma). *Gli1* (*Wen et al., 2010*) and *Ptch2* (*Somatilaka et al., 2020*) TaqMan probes for qRT-PCR were published before. Inventoried probes for *Gpr161*, and *Gapdh* were from Applied Biosystems. Reactions were run in CFX96 Real time System (Bio Rad).

## Gli1/2/3 immunoblotting

Embryos or limb buds were processed for Gli1/2/3 immunoblotting as described previously (*Wen et al., 2010*), using Gli3 (AF3690, R and D, 1 µg/ml), Gli2 (AF3635, R and D, 1 µg/ml), Gli1 (L42B10, Cell Signaling; 1:1000) and α-tubulin (clone DM1A, T6199, Sigma; 1:5000) antibodies. Secondary antibodies tagged with IRDye 680RD and IRDye 800CW for immunoblotting were from LI-COR Biosciences. Levels of Gli proteins were normalized to α-tubulin. Thirty µg of lysates were used in immunoblotting, well within the linear range of detection of α-tubulin levels using the DM1A monoclonal antibody and using LI-COR infrared laser-based detection.

## GPCR tandem affinity purifications

MEFs stably expressing LAP-tagged Gpr161 variants were lysed in buffer containing 50 mM Tris-HCl, pH 7.4, 200 mM KCl, 1 mM MgCl2, 1 mM EGTA, 10% glycerol, 1 mM DTT, 1% digitonin, 0.05% n-Dodecyl-β-D-Maltoside, 0.25% Cholesteryl hemisuccinate, 1 mM of AEBSF, 0.01 mg/mL of Leupeptin, pepstatin and chymostatin (*Pal et al., 2015*). Lysates were centrifuged at 12000xg for 10 min followed by tandem IPs (*Cheeseman and Desai, 2005*). Briefly, the GFP immunoprecipitates were first digested with TEV (N terminal LAP) or PreScission (for C terminal LAP) protease for 16 hr at 4°C. The supernatants were subjected to secondary IPs with S-protein agarose. The resulting secondary IPs were eluted in 2× urea sample buffer (4 M urea, 4% SDS, 100 mM Tris, pH 6.8, 0.2% bromophenol blue, 20% glycerol, and 200 mM DTT) at 37°C for 30 min and analyzed by immunoblotting (*Pal et al., 2015*). Tandem-IPs were run on 4–20% Mini-PROTEAN TGX Precast Protein Gels (Bio-Rad). Immunoblots from tandem affinity purifications were probed with antibodies against S-tag (mouse monoclonal MAC112; EMD Millipore) followed by visualization using IRDye-tagged secondary antibodies.

## In situ hybridization (ISH)

Antisense riboprobes were made using the following templates: *Ptch1, Gli1, Shh,* (gifts from Andrew McMahon lab and Deanna Grant, Andrew Peterson lab), *Sox9, Col2a1* (from Steven Vokes lab, UT Austin), *Hoxd13* (from Xin Sun lab, University of Wisconsin, Madison). Whole mount in situ

hybridization using digoxigenin-labeled probes was performed on embryos using standard protocols. Images were acquired using a Leica stereomicroscope (M165 C) with digital camera (DFC500) or Zeiss stereomicroscope (Discovery.V12) and AxioCam MRc.

## Quantification and statistical analysis

Cilia positive for acetylated $\alpha$-tubulin and GPR161/TULP3 were counted and expressed as % of GPR161/TULP3-positive cilia. To quantify ciliary pools of GPR161, TULP3, GFP, fluorescence levels were measured using the 'Measure' tool of Fiji software. Fluorescence levels of neighboring background areas were subtracted from that of the selected ciliary areas and expressed as corrected fluorescence. Quantification of dorsoventral extent of Nkx6.1 expression to that of the neural tube was done by averaging from two or more sections at thoracic region from two to four embryos. Statistical analyses were performed using Student's $t$-test for comparing two groups using GraphPad Prism.

## Acknowledgements

This paper is dedicated to the memory of Kathryn Anderson, who has been a constant source of encouragement and inspiration in our studies on ciliary signaling. This project was funded by Alex's Lemonade Foundation (A-grant to SM), and National Institutes of Health (1R01GM113023 to SM). We thank UT Southwestern transgenic and mouse animal care facility. We are thankful for kind gifts of reagents from Rajat Rohatgi, Robert Lefkowitz, Jonathan Eggenschwiler, Andrew McMahon, Deanna Grant, Andrew Peterson, Steven Vokes and Xin Sun. We thank Mukhopadhyay lab members for comments on the manuscript. Monoclonal antibodies developed by O D Madsen (Nkx6.1) and A Kawakami (Pax7) were obtained from the Developmental Studies Hybridoma Bank developed under the auspices of the NICHD and maintained by the Department of Biological Sciences, the University of Iowa, Iowa City IA 52242, USA.

## Additional information

### Funding

| Funder | Grant reference number | Author |
| --- | --- | --- |
| Alex's Lemonade Stand Foundation for Childhood Cancer | A-grant | Saikat Mukhopadhyay |
| National Institute of General Medical Sciences | 1R01GM113023 | Saikat Mukhopadhyay |

The funders had no role in study design, data collection and interpretation, or the decision to submit the work for publication.

### Author contributions

Sun-Hee Hwang, Conceptualization, Formal analysis, Validation, Investigation, Visualization, Methodology, Writing - review and editing; Bandarigoda N Somatilaka, Kevin White, Methodology; Saikat Mukhopadhyay, Conceptualization, Supervision, Funding acquisition, Writing - original draft, Project administration

### Author ORCIDs

Saikat Mukhopadhyay https://orcid.org/0000-0003-4790-3090

### Ethics

Animal experimentation: All the animals in the study were handled according to protocols approved by the UT Southwestern Institutional Animal Care and Use Committee, and the mouse colonies were maintained in a barrier facility at UT Southwestern, in agreement with the State of Texas legal and ethical standards of animal care. protocol 2016-101516.

**Decision letter and Author response**
Decision letter https://doi.org/10.7554/eLife.67121.sa1
Author response https://doi.org/10.7554/eLife.67121.sa2

# Additional files

## Supplementary files

- Transparent reporting form

## Data availability

All data generated or analyzed during this study are included in the manuscript and supporting files. All information on replicates is provided in figure legends. Most data are shown as scatter plots. In places where data means are provided, we have also added source data for Figure 4C, Figure 6D and Figure 4-supplement 1. No embryos or samples were excluded from analysis.

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

## Appendix 1

**Appendix 1—key resources table**

| Reagent type (species) or resource | Designation | Source or reference | Identifiers | Additional information |
|---|---|---|---|---|
| genetic reagent (*M. musculus*) | *Gpr161<sup>mut1/+</sup>* | This study | | See Methods |
| genetic reagent (*M. musculus*) | *Gpr161<sup>ko/+</sup>* | *Hwang et al., 2018* | RRID:MGI: 6357708 | |
| genetic reagent (*M. musculus*) | *Gpr161<sup>f/f</sup>* | *Hwang et al., 2018* | RRID:MGI: 6357710 | |
| genetic reagent (*M. musculus*) | *Wnt1-Cre* | Jackson Laboratory *Lewis et al., 2013* | Jax strain No: 022501 | |
| genetic reagent (*M. musculus*) | *Prx1-Cre* | Jackson Laboratory *Logan et al., 2002* | Jax strain No: 005584 | |
| genetic reagent (*M. musculus*) | *Gli2<sup>tm1Alj</sup>* | *Mo et al., 1997* | RRID:MGI: 1857509 | |
| cell line (*Homo sapiens*) | RPE hTERT | ATCC | ATCC CRL-4000; RRID:CVCL_4388 | |
| cell line (*M. musculus*) | NIH 3T3 Tet-on 3G | Clontech | 631197; RRID:CVCL_V360 | |
| cell line (*M. musculus*) | NIH-3T3 Flp-In | Thermo Fisher Scientific | R76107 | |
| cell line (*M. musculus*) | *Gpr161* ko NIH-3T3 Flp-In | *Pusapati et al., 2018* | | Gift from Rajat Rohatgi |
| cell line (*M. musculus*) | β-arrestin 1/2 double knockout MEFs | *Kovacs et al., 2008* | | Gift from R. Lefkowitz |
| transfected construct (*Homo sapiens*) | INPP5E On-target plus siRNA | Dharmacon/ Horizon, Perkin Elmer *Humbert et al., 2012* | J-020852–05 | 5'- GGAAUUA AAAGACGGAUUU-3' |
| transfected construct (*Homo sapiens*) | BBS4 siGenome siRNA | Dharmacon/ Horizon, Perkin Elmer *Loktev et al., 2008* | D-013649–04 | 5'- CGAUCUGACU UAUAUAAUG-3' |
| antibody | Mouse monoclonal anti-Acetylated tubulin | Sigma | Cat# T6793 RRID:AB_477585 | IF (1:5000) |
| antibody | Mouse monoclonal anti-Arl13b | NeuroMab Facility | Cat# N295B/66 RRID:AB_ 2750771 | IF (1:500) |
| antibody | Mouse monoclonal anti-γ-tubulin | Sigma | Cat# T6557 RRID:AB_532292 | IF (1:500) |
| antibody | Mouse monoclonal anti-α-tubulin | Sigma | Cat# T6199 RRID:AB_477583 | WB (1:5000) |
| antibody | Rabbit polyclonal anti-FoxA2 | Abcam | Cat# ab40874 RRID:AB_732411 or Cat# ab108422 RRID:AB_ 11157157 | IF (1:2000) |
| antibody | Mouse monoclonal anti-Nkx6.1 | DSHB | Cat# F55A10-s RRID:AB_532378 | IF (1:500) |

*Continued on next page*

*Appendix 1—key resources table continued*

| Reagent type (species) or resource | Designation | Source or reference | Identifiers | Additional information |
|---|---|---|---|---|
| antibody | Rabbit polyclonal anti-Pax6 | Biolegend | Cat# 901302 RRID:AB_2749901 | IF (1:50) |
| antibody | Mouse monoclonal anti-Pax7 | DSHB | RRID:AB_528428 | IF (1:50) |
| antibody | Guinea pig anti-Gli2 | Gift from Jonathan Eggenschwiler *Cho et al., 2008* | | IF (1:500) |
| antibody | Chicken anti-GFP | Abcam | Cat# ab13970 RRID:AB_300798 | IF (1:10000) |
| antibody | Rabbit polyclonal anti-Gpr161 | Custom-made *Pal et al., 2016* | | IF (1:250) |
| antibody | Rabbit polyclonal anti-Tulp3 | Gift from Jonathan Eggenschwiler *Norman et al., 2009* | | IF (1:500) |
| antibody | Goat polyclonal anti-Gli3 | R and D Systems | AF3690 | WB (1:1000) |
| antibody | Goat polyclonal anti-Gli2 | R and D Systems | AF3635 | WB (1:1000) |
| antibody | Mouse monoclonal anti-Gli1 | Cell Signaling RRID:AB_2294746 | Cat# 2643 | WB (1:1000) |
| antibody | Alexa Fluor 488-, 555-, 594-, 647- conjugated secondary antibodies | Life Technologies, Carlsbad, CA | | IF (1:2000) |
| antibody | IRDye tagged secondary antibodies | LI-COR | | IF (1:5000) |
| antibody | Hoechst 33342 | Life Technologies | | IF (1:5000) |
| recombinant DNA reagent | pgLAP5 vector (plasmid) | Addgene | Plasmid #19706; RRID:Addgene_19706 | |
| recombinant DNA reagent | pDONR221_Gpr161 (*M. musculus*) | *Mukhopadhyay et al., 2013* | Synthesized by DNA2.0 | NM_001081126.1 |
| sequence-based reagent | *Gpr161 mut1* F | This paper | PCR primers | CAG AAA GCA ACA GCA AAG CA |
| sequence-based reagent | *Gpr161 mut1* R | This paper | PCR primers | ACC CTG ACA CTG CCC TTA GC |
| sequence-based reagent | *Gpr161* Ex4 F | *Hwang et al., 2018* | PCR primers | CAA GAT GGA TTC GCA GTA GCT TGG |
| sequence-based reagent | *Gpr161* floxed R | *Hwang et al., 2018* | PCR primers | ATG GGG TAC ACC ATT GGA TAC AGG |
| sequence-based reagent | *Gpr161 Neo cassette* R | *Hwang et al., 2018* | PCR primers | CAA CGG GTT CTT CTG TTA GTC C |
| sequence-based reagent | Cre-F | *Hwang et al., 2018* | PCR primers | AAT GCT GTC ACT TGG TCG TGG C |
| sequence-based reagent | Cre-R | *Hwang et al., 2018* | PCR primers | GAA AAT GCT TCT GTC CGT TTG C |
| sequence-based reagent | Gli2 sense | *Mo et al., 1997* | PCR primers | AAA CAA AGC TCC TGT ACA CG |

*Continued on next page*

*Appendix 1—key resources table continued*

| Reagent type (species) or resource | Designation | Source or reference | Identifiers | Additional information |
|---|---|---|---|---|
| sequence-based reagent | Gli2 antisense | *Mo et al., 1997* | PCR primers | CAC CCC AAA GCA TGT GTT TT |
| sequence-based reagent | pPNT | *Mo et al., 1997* | PCR primers | ATG CCT GCT CTT TAC TGA AG |
| sequence-based reagent | Gli1-F | Sigma | *Gli1* TaqMan probes for qRT-PCR | GCA GTG GGT AAC ATG AGT GTC T |
| sequence-based reagent | Gli1-R | Sigma | *Gli1* TaqMan probes for qRT-PCR | AGG CAC TAG AGT TGA GGA ATT GT |
| sequence-based reagent | Gli1-Probe (FAM/TAMRA) | Sigma | *Gli1* TaqMan probes for qRT-PCR | CTC TCC AGG CAG AGA CCC AG C |
| sequence-based reagent | Ptch2-F | Sigma | *Ptch2* TaqMan probes for qRT-PCR | CAG AGT GAC TAC CTC CAT GAC TGT |
| sequence-based reagent | Ptch2-R | Sigma | *Ptch2* TaqMan probes for qRT-PCR | GCT GGG TGG ACG TAT GCT |
| sequence-based reagent | Ptch2-probe (FAM/TAMRA) | Sigma | *Ptch2* TaqMan probes for qRT-PCR | CTC CAC CCA CCA CCT CTG CCT |
| sequence-based reagent | Taqman probes for *Gpr161* | Applied Biosystems | Mm01291057_m1 | |
| sequence-based reagent | Taqman probes for *Gapdh* | Applied Biosystems | Mm99999915 | |
| commercial assay or kit | TR-FRET cAMP assay | Cisbio | 62AM4PEB | |
| commercial assay or kit | GenElute mammalian total RNA purification kit | Sigma | RTN350 | |
| commercial assay or kit | SYBR Green Quantitative RT-qPCR Kit | Sigma | QR0100 | |
| commercial assay or kit | Mycoplasma PCR Detection Kit | Genlantis | MY01050 | |
| commercial assay or kit | Kicqstart One-Step Probe RT-qPCR ReadyMix | Sigma | KCQS07 | |
| chemical compound, drug | Penicillin/Streptomycin | Sigma | P4333 | Cell culture |
| chemical compound, drug | Glutamine | Sigma | G7513 | Cell culture |
| chemical compound, drug | Polyfect | Qiagen | 301105 | Cell culture |
| chemical compound, drug | DMEM-High Glucose | Sigma | D5796 | Cell culture |

*Continued on next page*

*Appendix 1—key resources table continued*

| Reagent type (species) or resource | Designation | Source or reference | Identifiers | Additional information |
|---|---|---|---|---|
| chemical compound, drug | DMEM F12 | Sigma | D6421 | Cell culture |
| chemical compound, drug | BCS | Sigma | 12133C | Cell culture |
| chemical compound, drug | FBS | Sigma | F0926 | Cell culture |
| chemical compound, drug | Paraformaldehyde | Electron microscopy solutions | 15710 | |
| chemical compound, drug | Normal donkey serum | Jackson Immuno Research, West Grove, PA | Cat# 017-000-121; RRID:AB_2337258 | |
| chemical compound, drug | Fluoromount-G | Southern Biotech | 0100–01 | |
| chemical compound, drug | Permount | ThermoFisher Scientific | SP15-100 | |
| software, algorithm | ImageJ software | | RRID:SCR_003070 | |
| software, algorithm | GraphPad Prism | | RRID:SCR_002798 | |

