## [Decision Letter]

**Acceptance summary:**

This paper will be of interest to scientists interested in GPR161 biology and its ciliary regulation of Hedgehog (Hh) signaling. The phenotypes observed in a new GPR161 mutant mouse carrying a hypomorphic allele provide additional information about which developing tissues are more sensitive to Gpr161 function.

**Decision letter after peer review:**

Thank you for submitting your article "Deficient ciliary Gpr161 pools cause high hedgehog signaling phenotypes by reducing Gli repression but not by activation" for consideration by *eLife*. Your article has been reviewed by 2 peer reviewers, and the evaluation has been overseen by a Reviewing Editor and Suzanne Pfeffer as the Senior Editor. The following individual involved in review of your submission has agreed to reveal their identity: Maxence V Nachury (Reviewer #2).

As you will see from the detailed reviewer comments, the referees did not think that the data at this stage support the model that the ciliary role of Gpr161 is related to its effect on Gli3R, while its extra-ciliary role is related to regulation of GliA. Measurement of ciliary, peri-ciliary and cytoplasmic cAMP levels would help but this is not a standard experiment. They wrote, "One way to make the paper work is to moderate the claims and focus on the conclusion that "Gpr161 suppresses Hh signaling through both ciliary and non-ciliary mechanisms". The authors should focus on the mouse phenotypes without linking these two mechanisms to specific biochemical (ie. Gli3R and GliA) or cell biological (ie. ciliary and extra-ciliary cAMP) processes (for which they provide little/no experimental support)." In addition, one wrote, "The authors should try a few simple experiments to strengthen the basic premise of the paper, that GPR161mut1 is truly never in cilia. Depleting β arrestin and/or the BBSome in their MEFs will give a rapid answer to whether this mutant exits cilia more rapidly than the WT."

Reviewer #1 (Recommendations for the authors):

All recommendations are in public review.

Reviewer #2 (Recommendations for the authors):

Quantitation of WB signals (Figure 4C). Are signals within the linear range of measurement? While manufacturers often claim instrument linearity within 4 or 5 orders of magnitudes, empiric validation often finds these claims to be exaggerated. A simple immunoblot with varying amounts of samples loaded would alleviate the concern of non-linear response.

Figure 4C: The mut1/mut1 data on the Gli3FL/R ratio do not support the statement 'the Gpr161 mutants exhibit gradually decreasing severity of Gli3 processing defects compared to ko, commensurate with monoallelic or biallelic expression of Gpr161mut1.' The Gli3FL/R ratio is lower in mut1/mut1 than in any other allele combination, including WT. Why does the mut1/mut1 allele combination cause an increased production of Gli3R relative to the amount of Gli3FL? It is interesting that the signaling defects of the mut1/mut1 animals are extremely modest. One interpretation is that GPR161 ciliary localization plays no detectable role at E9.5. Another interpretation is that GPR161mut1 does localize to cilia in cells (e.g. from the neural tube) of E9.5 embryos. The authors need to discuss the reduction of the Gli3FL/R ratio in mut1/mut1 animals (Figure 4C) and may wish to stain E9.5 embryos to reject the hypothesis that GPR161mut1 localizes to cilia at this stage of development.

On the other hand, the production of Gli3R is clearly compromised in the mut1/mut1 mouse limbs at E12.5. It is comforting that the morphological and molecular phenotypes are congruent.

Figure 8 is helpful in orienting the reader through a complex set of data. The authors could add the 'no GPR161' situation to echo the discussion. In the same vein, adding diagrams showing the author's thinking of how ciliary vs. non ciliary pools of GPR161 are affecting Gli3R and Gli2A production would be helpful. The point that I have the hardest time reconciling with the presented data is that cilia are required to generate both activator and repressor forms of Gli. Yet, extra ciliary pools of GPR161 play a role in generating GLI2A.

Some text constructions are ambiguous or grammatically incorrect. A few examples are given below but the list is not exhaustive. In general, the text would benefit from careful editing.

'High hedgehog signaling phenotypes in Gpr161 knockout occur from Gli-repressor lack or Gli-activator generation.'

'Here we demonstrate distinctive roles for ciliary Gpr161 pools in preventing hedgehog hyperactivation phenotypes, contingent upon dependence of the morpho-phenotypic spectrum on downstream Gli repression'

'Ventral expansion in Ankmy2 knockout is severe than Gpr161 knockout, but even then, is fully *Gli2*-dependent'

'Gpr161 deletion showed GliR lack'

'However, lack of Gpr161 in cilia was incapable of sufficient Gli-activator generation required in causing ventral-most progenitor expansion in the neural tube.'

'Gpr161 ciliary pools regulate Hh repression regulated morpho-phenotypic spectrum'

Typo

'Gpr16mut/mut11 embryos exhibit high Hh signaling 1 and polydactyly in limb buds'

Some inaccurate statements of facts:

'Gpr161 C-tail has been proposed to be an A-kinase anchoring protein (AKAP) for PKA activation in cilia by binding to PKA regulatory subunit RIa and RIb (Bachmann et al. 2016).'

'Direct Gpr161 C-tail binding to PKA RIa/RIb regulatory subunits (Bachmann et al. 2016) facilitates PKA activation in close proximity to Gpr161-mediated cAMP production in cilia for GliR processing.' The Bachmann reference shows that the AKAP motif of GPR161 is specific for PKA RIalpha

'Agonist-independent constitutive signaling.' The author may want to leave open the possibility that the tonic activity of GPR161 is caused by an agonist endogenous to the experimental system.

'protein kinase A (PKA)-mediated limited proteolysis of full length *Gli2*/3 into *Gli2*/3 repressor (GliR) forms.' PKA does not mediate proteolysis

'Increased extracellular Gpr161 pools in Gpr161mut1/mut1 also prevent Nkx6.1 ventralization from likely restoration of Gli3R processing.' Extra-ciliary?

[Editors' note: further revisions were suggested prior to acceptance, as described below.]

Thank you for resubmitting your work entitled "Ciliary and extraciliary Gpr161 pools repress hedgehog signaling in a tissue-specific manner" for further consideration by *eLife*. Your article has been reviewed by 2 peer reviewers, and the evaluation has been overseen by a Reviewing Editor and Suzanne Pfeffer as the Senior Editor.

The manuscript has been improved but there are some remaining issues that need to be addressed, as outlined below.

The reviewers felt that dissecting the ciliary vs. non-ciliary role of this GPCR is an important advance. However, as discussed below, it will be necessary to revise both the abstract and conclusion to explicitly acknowledge that you have not shown that their mutant Gpr161 changes ciliary cAMP levels and that you have not demonstrated conclusively that the phenotypes seen in these mutants are due to loss of Gli3 repressor. One reviewer wrote, "The text of the paper still leaves one with the (erroneous) impression that they have altered "cAMP signaling". I think all they can say is that they have identified a mutant of Gpr161 that has reduced localization in cilia and the phenotypes caused by this mutant are different from the null (depending on the tissue examined). I think they should add a separate paragraph to the conclusion highlighting these limitations."

Please prepare a carefully revised version that addresses these concerns:

Reviewer #1:

I appreciate the effort the authors have made to improve data presentation of phenotypes, clarify localization of Gpr161 and discuss the recent paper in zebrafish showing the importance of ciliary cAMP pools for Hedgehog signaling.

However, the authors have not addressed two major issues raised in my previous review:

1. Does the mutation in Gpr161 (or its loss) cause a change in the ciliary levels of cAMP ? While there is an extensive discussion of the ciliary cAMP pools throughout the paper (and even a mention of this in the last sentence of the abstract), a demonstration that Gpr161 mutations changes the cAMP pool in cilia is lacking. Without additional data, the text must be carefully toned down as not to mislead the reader.

2. Again, while there is extensive discussion in this area, the authors do not show that Gpr161 mutation causes phenotypes in some tissues that are Gli3 repressor dependent. A simple experiment to address this is to show that the phenotypes in question can be rescued by re-expression of Gli3R. This would establish cause-effect rather than correlation. Without this, the limitations of the study must be stated clearly and conclusions softened.

Reviewer #2:

A major point raised in the first round of review was the limited evidence that GPR161mut1 fails to enter cilia. The added data considerably strengthen this point. GPR161mut1 is still absent from cilia in the exit mutants Arrb1/Arrb2 DKO and in cells depleted of BBS4 by siRNA.

In addition, the authors now show that GPR161mut1 is present in pericentriolar recycling endosomes, a notable point that enriches the discussion. This observation introduces the possibility that percentriolar cAMP generated by GPR161mut1 finds its way into cilia to raise ciliary cAMP levels to a level that suppresses formation of GliA.

The text has been modified to temper some of the conclusions and more accurately describe the results. However, there are many instances where the writing is too convoluted and lacks clarity. In particular, the extensive re-writing of the abstract has made for some rather complex statements that only a few specialists will be able to comprehend. The level of textual editing required falls outside of the scope of peer-review. I had listed a few instances in my initial review but indicated that the list was not exhaustive. Yet, it appears that the authors only addressed the instances listed. It is a pity that clarity is still lacking as the lasting impact of this paper would be diminished if the paper were published in its current state. The authors may seek to engage the service of a professional editor.

One instance is listed below but there are many more.

'We conclusively ruled out that LAP-tagged Gpr161mut1 (LAP, S tag-PreScission-GFP) does not at all transit through cilia'. What do the authors rule out?

---

## [Author Response]

The reviewers have discussed their reviews with one another, and the Reviewing Editor has drafted this to help you prepare a revised submission.As you will see from the detailed reviewer comments, the referees did not think that the data at this stage support the model that the ciliary role of Gpr161 is related to its effect on Gli3R, while its extra-ciliary role is related to regulation of GliA. Measurement of ciliary, peri-ciliary and cytoplasmic cAMP levels would help but this is not a standard experiment. They wrote, "One way to make the paper work is to moderate the claims and focus on the conclusion that "Gpr161 suppresses Hh signaling through both ciliary and non-ciliary mechanisms". The authors should focus on the mouse phenotypes without linking these two mechanisms to specific biochemical (ie. Gli3R and GliA) or cell biological (ie. ciliary and extra-ciliary cAMP) processes (for which they provide little/no experimental support)."

We have revised the title of the paper and the focus of our discussion based on these suggestions. We have also discussed the limitations of our Gpr161^mut1^ mouse model and other optogenetic/chemogenetic methods for generating cAMP in cilia. These limitations arise from the cilioplasm not being strictly restricted from the cytoplasm. Therefore, the second messengers cAMP and ca^2+^ are freely diffusible between ciliary and extraciliary compartments (Delling et al., 2016; Truong et al., 2021). A paper published in Cell during revision of this study used optogenetic tools to show that ciliary, but not cytoplasmic, production of cAMP functions through PKA localized in cilia (Truong et al., 2021) to repress sonic hedgehog-mediated somite patterning in zebrafish (Wolff et al., 2003). We have also compared and discussed these results with our study. Our study highlights that the effects of ciliary loss of Gpr161 pools are tissue specific and dependent on the requirements of the tissues on GliR vs GliA in the morpho-phenotypic spectrum. Overall, our results using Gpr161^mut1^ allele are strikingly complementary to the optogenetic study by showing that lack of ciliary Gpr161 pools result in Hh hyperactivation phenotypes arising mainly from lack of GliR, in the limb buds, mid-face and intermediate neural tube.

In addition, one wrote, "The authors should try a few simple experiments to strengthen the basic premise of the paper, that GPR161mut1 is truly never in cilia. Depleting β arrestin and/or the BBSome in their MEFs will give a rapid answer to whether this mutant exits cilia more rapidly than the WT."

We have now shown that Gpr161^mut1^ is not present in cilia in β-arrestin1/2 (Arrb1/2) double ko MEFs or upon RNAi of BBS4. We already showed that knockdown of the 5’phosphpatase INPP5E that causes accumulation of Gpr161 in cilia does not show any accumulation of Gpr161^mut1^ in cilia. Based on all these experiments, we surmise that Gpr161^mut1^ does not transit through cilia.

Reviewer #2 (Recommendations for the authors):Quantitation of WB signals (Figure 4C). Are signals within the linear range of measurement? While manufacturers often claim instrument linearity within 4 or 5 orders of magnitudes, empiric validation often finds these claims to be exaggerated. A simple immunoblot with varying amounts of samples loaded would alleviate the concern of non-linear response.

Levels of Gli proteins in Figure 4 were normalized to α-tubulin. 30 µg of mouse embryo lysates were used in immunoblotting, well within the linear range of detection of α-tubulin levels using the DM1A monoclonal antibody and LI-COR infrared laser-based detection (please see Author response image 1). We would also like to note that IRDye detection is quantitative over a much broader linear dynamic range than ECL. Fluorescent conjugates avoid the enzyme kinetics and substrate availability limitations of chemiluminescence. This produces more consistent, straightforward, and accurate quantification (Gerk, 2011; Wang et al., 2007).

**Author response image 1. sa2fig1:** Quantification of α-Tubulin intensities using LI-COR infrared laser-based detection as a function of loaded lysate amount.

Figure 4C: The mut1/mut1 data on the Gli3FL/R ratio do not support the statement 'the Gpr161 mutants exhibit gradually decreasing severity of Gli3 processing defects compared to ko, commensurate with monoallelic or biallelic expression of Gpr161mut1.' The Gli3FL/R ratio is lower in mut1/mut1 than in any other allele combination, including WT. Why does the mut1/mut1 allele combination cause an increased production of Gli3R relative to the amount of Gli3FL? It is interesting that the signaling defects of the mut1/mut1 animals are extremely modest. One interpretation is that GPR161 ciliary localization plays no detectable role at E9.5. Another interpretation is that GPR161mut1 does localize to cilia in cells (e.g. from the neural tube) of E9.5 embryos. The authors need to discuss the reduction of the Gli3FL/R ratio in mut1/mut1 animals (Figure 4C) and may wish to stain E9.5 embryos to reject the hypothesis that GPR161mut1 localizes to cilia at this stage of development.On the other hand, the production of Gli3R is clearly compromised in the mut1/mut1 mouse limbs at E12.5. It is comforting that the morphological and molecular phenotypes are congruent.

As the reviewer suggests, intermediate level neural tube ventralization manifested from loss of ciliary Gpr161 but was suppressed by increased extraciliary Gpr161^mut1^ generating Gli3-repressor. Morphogenesis in limb buds and midface require Gli-repressor. These tissues in Gpr161^mut1^ manifested hedgehog hyperactivation phenotypes—polydactyly and midfacial widening. We think that compartmentalized ciliary cAMP signaling establishes tissue-specific repression thresholds in dictating these distinct morpho-phenotypic outcomes. We have discussed the differences in GliR processing between tissues and also in the morpho-phenotypic outcomes in the updated Discussion.

Figure 8 is helpful in orienting the reader through a complex set of data. The authors could add the 'no GPR161' situation to echo the discussion. In the same vein, adding diagrams showing the author's thinking of how ciliary vs. non ciliary pools of GPR161 are affecting Gli3R and Gli2A production would be helpful. The point that I have the hardest time reconciling with the presented data is that cilia are required to generate both activator and repressor forms of Gli. Yet, extra ciliary pools of GPR161 play a role in generating GLI2A.

Thanks for the suggestions. First, we have added a “no Gpr161” vs “no Gpr161 in cilia” comparison of phenotypes in revised Figure 8A-C. Second, we have also added a new diagram (Figure 8D) showing how we think the ciliary and extraciliary pools of Gpr161 are affecting Gli3R and Gli2A production.

Some text constructions are ambiguous or grammatically incorrect. A few examples are given below but the list is not exhaustive. In general, the text would benefit from careful editing.

Thank you for all the corrections. We have revised the text accordingly.

'High hedgehog signaling phenotypes in Gpr161 knockout occur from Gli-repressor lack or Gli-activator generation.''Here we demonstrate distinctive roles for ciliary Gpr161 pools in preventing hedgehog hyperactivation phenotypes, contingent upon dependence of the morpho-phenotypic spectrum on downstream Gli repression''Ventral expansion in Ankmy2 knockout is severe than Gpr161 knockout, but even then, is fully Gli2-dependent''Gpr161 deletion showed GliR lack''However, lack of Gpr161 in cilia was incapable of sufficient Gli-activator generation required in causing ventral-most progenitor expansion in the neural tube.''Gpr161 ciliary pools regulate Hh repression regulated morpho-phenotypic spectrum'Typo'Gpr16mut/mut11 embryos exhibit high Hh signaling 1 and polydactyly in limb buds'Some inaccurate statements of facts:'Gpr161 C-tail has been proposed to be an A-kinase anchoring protein (AKAP) for PKA activation in cilia by binding to PKA regulatory subunit RIa and RIb (Bachmann et al. 2016).'

The Bachmann paper, although focused on RIα, does show that Gpr161 C-tail binds to PKA regulatory subunit RIβ . They also find corresponding decrease in binding in a neurodegenerative patient mutation in RIβ (L50D) (please see Figure 3D in (Bachmann et al., 2016)). We do realize that much of the paper discussed RIα interactions. We have thus simplified the text throughout to reflect Gpr161 C-tail interactions with type I PKA regulatory subunits.

'Direct Gpr161 C-tail binding to PKA RIa/RIb regulatory subunits (Bachmann et al. 2016) facilitates PKA activation in close proximity to Gpr161-mediated cAMP production in cilia for GliR processing.' The Bachmann reference shows that the AKAP motif of GPR161 is specific for PKA RIalpha

Please see previous comment.

'Agonist-independent constitutive signaling.' The author may want to leave open the possibility that the tonic activity of GPR161 is caused by an agonist endogenous to the experimental system.

Thanks, we now discuss this specific limitation in Discussion.

'protein kinase A (PKA)-mediated limited proteolysis of full length Gli2/3 into Gli2/3 repressor (GliR) forms.' PKA does not mediate proteolysis

Thanks, corrected.

'Increased extracellular Gpr161 pools in Gpr161mut1/mut1 also prevent Nkx6.1 ventralization from likely restoration of Gli3R processing.' Extra-ciliary?

Thanks, corrected.

References

Bachmann, V.A., Mayrhofer, J.E., Ilouz, R., Tschaikner, P., Raffeiner, P., Rock, R., Courcelles, M., Apelt, F., Lu, T.W., Baillie, G.S.*, et al.* (2016). Gpr161 anchoring of PKA consolidates GPCR and cAMP signaling. Proc Natl Acad Sci U S A *113*, 7786-7791.

Bock, A., Annibale, P., Konrad, C., Hannawacker, A., Anton, S.E., Maiellaro, I., Zabel, U., Sivaramakrishnan, S., Falcke, M., and Lohse, M.J. (2020). Optical Mapping of cAMP Signaling at the Nanometer Scale. Cell *182*, 1519-1530 e1517.

Delling, M., Indzhykulian, A.A., Liu, X., Li, Y., Xie, T., Corey, D.P., and Clapham, D.E. (2016). Primary cilia are not calcium-responsive mechanosensors. Nature *531*, 656-660.

Gerk, P.M. (2011). Quantitative immunofluorescent blotting of the multidrug resistance-associated protein 2 (MRP2). J Pharmacol Toxicol Methods *63*, 279-282.

Guo, J., Otis, J.M., Suciu, S.K., Catalano, C., Xing, L., Constable, S., Wachten, D., Gupton, S., Lee, J., Lee, A.*, et al.* (2019). Primary Cilia Signaling Promotes Axonal Tract Development and Is Disrupted in Joubert Syndrome-Related Disorders Models. Dev Cell *51*, 759-774 e755.

Hansen, J.N., Kaiser, F., Klausen, C., Stuven, B., Chong, R., Bonigk, W., Mick, D.U., Moglich, A., Jurisch-Yaksi, N., Schmidt, F.I.*, et al.* (2020). Nanobody-directed targeting of optogenetic tools to study signaling in the primary cilium. *eLife 9*.

Mukhopadhyay, S., Wen, X., Ratti, N., Loktev, A., Rangell, L., Scales, S.J., and Jackson, P.K. (2013). The ciliary G-protein-coupled receptor Gpr161 negatively regulates the Sonic hedgehog pathway via cAMP signaling. Cell *152*, 210-223.

Truong, M.E., Bilekova, S., Choksi, S.P., Li, W., Bugaj, L.J., Xu, K., and Reiter, J.F. (2021). Vertebrate cells differentially interpret ciliary and extraciliary cAMP. Cell.

Wang, Y.V., Wade, M., Wong, E., Li, Y.C., Rodewald, L.W., and Wahl, G.M. (2007). Quantitative analyses reveal the importance of regulated Hdmx degradation for p53 activation. Proc Natl Acad Sci U S A *104*, 12365-12370.

Wolff, C., Roy, S., and Ingham, P.W. (2003). Multiple muscle cell identities induced by distinct levels and timing of hedgehog activity in the zebrafish embryo. Curr Biol *13*, 1169-1181.

Zhang, J.Z., Lu, T.W., Stolerman, L.M., Tenner, B., Yang, J.R., Zhang, J.F., Falcke, M., Rangamani, P., Taylor, S.S., Mehta, S.*, et al.* (2020). Phase Separation of a PKA Regulatory Subunit Controls cAMP Compartmentation and Oncogenic Signaling. Cell *182*, 1531-1544 e1515.

[Editors' note: further revisions were suggested prior to acceptance, as described below.]

The reviewers felt that dissecting the ciliary vs. non-ciliary role of this GPCR is an important advance. However, as discussed below, it will be necessary to revise both the abstract and conclusion to explicitly acknowledge that you have not shown that their mutant Gpr161 changes ciliary cAMP levels and that you have not demonstrated conclusively that the phenotypes seen in these mutants are due to loss of Gli3 repressor. One reviewer wrote, "The text of the paper still leaves one with the (erroneous) impression that they have altered "cAMP signaling". I think all they can say is that they have identified a mutant of Gpr161 that has reduced localization in cilia and the phenotypes caused by this mutant are different from the null (depending on the tissue examined). I think they should add a separate paragraph to the conclusion highlighting these limitations."

We have extensively revised the abstract and the discussion explicitly addressing the above concerns. We have removed any reference to Gpr161^mut1^ functioning through cAMP signaling in cilia in the abstract. We have also added a *separate* section in Discussion highlighting the limitations of the current *Gpr161* allele termed as “Limitations of the *Gpr161^mut1^* allele.”

This section explicitly addresses the limitations of *Gpr161^mut1^* allele with regards to our lack of understanding of its role in ligand-induced cAMP signaling, ciliary cAMP signaling, and through loss of Gli3 repressor formation.

For e.g., here we mention that “Currently, a limitation of the *Gpr161^mut1^
*allele arises from the fact that a physiological ligand, if any, for Gpr161 is unknown. Thus, we are unable to test role of Gpr161^mut1^ in ligand-induced cAMP signaling and formally rule out if this mutant is functioning as a hypomorph, in addition to being extraciliary.”

We further mention that “In addition, there have been conflicting reports of ciliary cAMP concentration in kidney epithelial cells ranging from levels comparable to that in the cytoplasm (Jiang et al., 2019) to supraphysiological levels ~4.5 µM (Moore et al., 2016). In the absence of a known ligand for Gpr161, and lack of consensus regarding efficacy of the currently available ciliary targeted cAMP probes, we are unable to show impact of Gpr161^mut1^ on ciliary cAMP levels.”

We also mention that:

“Currently, we have not demonstrated conclusively that the resultant phenotypes in *Gpr161^mut1^* are from loss of Gli3 repressor. The commonly used *Gli3R* allele (*Gli3∆699*) expresses a fusion protein containing the first 699 aa of Gli3 and extra residues from thymidine kinase used for generating the allele (Bose et al., 2002). The *Gli3∆699* allele is of limited potency as a Gli3R, because it does not resemble *Shh* ko during limb development (Cao et al., 2013) as would be expected from Gli3R antagonizing Shh activity (Litingtung et al., 2002; Persson et al., 2002). Introducing improved *Gli3R* alleles with potency equal to that of the natural repressor (*Gli3∆701*) (Cao et al., 2013) in the background of *Gpr161* and/or *Gli2* mutants will be required in the future to conclusively show role of GliR in *Gpr161^ko^/Gpr161^mut1^* phenotypes.”

While we show that Gpr161^mut1^ is cAMP signaling competent using expression in non-ciliated cells and TR-FRET assays measuring total cellular cAMP, measuring cAMP levels in cilia is thus technically limited by lack of any known ligand for Gpr161 and available probes. However, there is growing consensus in the field that Gpr161 functions through Gαs-cAMP signaling, PKA activation and GliR formation, at least in the context of developmental pathways (Hwang et al., 2018; Shimada et al., 2018; Shimada et al., 2019) and across the vertebrate lineage (Bachmann et al., 2016; May et al., 2021; Mukhopadhyay et al., 2013; Tschaikner et al., 2021).

Please prepare a carefully revised version that addresses these concerns:Reviewer #1:I appreciate the effort the authors have made to improve data presentation of phenotypes, clarify localization of Gpr161 and discuss the recent paper in zebrafish showing the importance of ciliary cAMP pools for Hedgehog signaling.

Thank you.

However, the authors have not addressed two major issues raised in my previous review:1. Does the mutation in Gpr161 (or its loss) cause a change in the ciliary levels of cAMP ? While there is an extensive discussion of the ciliary cAMP pools throughout the paper (and even a mention of this in the last sentence of the abstract), a demonstration that Gpr161 mutations changes the cAMP pool in cilia is lacking. Without additional data, the text must be carefully toned down as not to mislead the reader.

We have now added a separate section in Discussion addressing the limitations of *Gpr161^mut1^* allele with regards to our lack of understanding of its role in ligand-induced cAMP signaling. We have also removed any reference to Gpr161^mut1^ functioning through cAMP signaling in cilia in the abstract. Please note that there have been conflicting reports of ciliary cAMP concentration ranging from levels comparable to that in the cytoplasm (Jiang et al., 2019) to supraphysiological levels ~4.5 µM (Moore et al., 2016). Thus, clearly, the cAMP probes need further optimization. While we show that Gpr161^mut1^ is cAMP signaling competent using expression in non-ciliated cells, measuring cAMP levels in cilia is technically limited by lack of any known ligand for Gpr161.

2. Again, while there is extensive discussion in this area, the authors do not show that Gpr161 mutation causes phenotypes in some tissues that are Gli3 repressor dependent. A simple experiment to address this is to show that the phenotypes in question can be rescued by re-expression of Gli3R. This would establish cause-effect rather than correlation. Without this, the limitations of the study must be stated clearly and conclusions softened.

We agree with the reviewer that we have not demonstrated conclusively that the resultant phenotypes in *Gpr161^mut1^* are from loss of Gli3 repressor. Further demonstration would require introducing improved *Gli3R* alleles with potency equal to the natural repressor (Cao et al., 2013) in the background of *Gpr161/Gpr161^mut1^* and/or *Gli2* loss. Please note that the commonly used *Gli3R* allele (*Gli3∆699*) expresses a fusion protein containing the first 699 aa of Gli3 and extra 21 residues from the thymidine kinase gene sequence used for generating the mutant allele (Bose et al., 2002). This allele does not resemble *Shh* ko during limb development (Cao et al., 2013), as expected from Gli3R antagonizing Shh activity (Litingtung et al., 2002; Persson et al., 2002), suggesting limited potency as a Gli3R. These experiments using *Gli3R* alleles with potency equal to the natural repressor (*Gli3∆701*) (Cao et al., 2013) in the background of *Gpr161^ko^/Gpr161^mut1^* and/or *Gli2* loss are currently beyond the scope of the current manuscript. In addition, *Gpr161* and *Gli2* are on the same chromosome, so generating a double mutant for *Gpr161^mut1^* and *Gli2* knockout will require linking through genetic recombination by breeding *Gpr161^mut1^
*with *Gli2^ko/+^
*animals, as described for *Gpr161* floxed and *Gli2* knockout alleles in the Methods. We have discussed these limitations in the current manuscript in the separate section in Discussion on limitations of the *Gpr161^mut1^* allele.

Reviewer #2:A major point raised in the first round of review was the limited evidence that GPR161mut1 fails to enter cilia. The added data considerably strengthen this point. GPR161mut1 is still absent from cilia in the exit mutants Arrb1/Arrb2 DKO and in cells depleted of BBS4 by siRNA.

Thank you.

In addition, the authors now show that GPR161mut1 is present in pericentriolar recycling endosomes, a notable point that enriches the discussion. This observation introduces the possibility that percentriolar cAMP generated by GPR161mut1 finds its way into cilia to raise ciliary cAMP levels to a level that suppresses formation of GliA.

Thank you.

The text has been modified to temper some of the conclusions and more accurately describe the results.

Thank you.

However, there are many instances where the writing is too convoluted and lacks clarity. In particular, the extensive re-writing of the abstract has made for some rather complex statements that only a few specialists will be able to comprehend. The level of textual editing required falls outside of the scope of peer-review. I had listed a few instances in my initial review but indicated that the list was not exhaustive. Yet, it appears that the authors only addressed the instances listed. It is a pity that clarity is still lacking as the lasting impact of this paper would be diminished if the paper were published in its current state. The authors may seek to engage the service of a professional editor.

Our apologies if the text seemed complex. We have now carefully revised the text throughout the Abstract and Discussion.

One instance is listed below but there are many more.'We conclusively ruled out that LAP-tagged Gpr161mut1 (LAP, S tag-PreScission-GFP) does not at all transit through cilia'. What do the authors rule out?

Thank you for pointing to this gaffe and we are embarrassed by the mistake. We have corrected the statement as follows:

“These experiments conclusively demonstrate that Gpr161^mut1^ does not at all transit through cilia.”

References

Bachmann, V.A., Mayrhofer, J.E., Ilouz, R., Tschaikner, P., Raffeiner, P., Rock, R., Courcelles, M., Apelt, F., Lu, T.W., Baillie, G.S.*, et al.* (2016). Gpr161 anchoring of PKA consolidates GPCR and cAMP signaling. Proc Natl Acad Sci U S A *113*, 7786-7791.

Bose, J., Grotewold, L., and Ruther, U. (2002). Pallister-Hall syndrome phenotype in mice mutant for Gli3. Hum Mol Genet *11*, 1129-1135.

Cao, T., Wang, C., Yang, M., Wu, C., and Wang, B. (2013). Mouse limbs expressing only the Gli3 repressor resemble those of Sonic hedgehog mutants. Dev Biol *379*, 221-228.

Hwang, S.H., White, K.A., Somatilaka, B.N., Shelton, J.M., Richardson, J.A., and Mukhopadhyay, S. (2018). The G protein-coupled receptor Gpr161 regulates forelimb formation, limb patterning and skeletal morphogenesis in a primary cilium-dependent manner. Development *145*.

Jiang, J.Y., Falcone, J.L., Curci, S., and Hofer, A.M. (2019). Direct visualization of cAMP signaling in primary cilia reveals up-regulation of ciliary GPCR activity following Hedgehog activation. Proc Natl Acad Sci U S A *116*, 12066-12071.

Litingtung, Y., Dahn, R.D., Li, Y., Fallon, J.F., and Chiang, C. (2002). Shh and Gli3 are dispensable for limb skeleton formation but regulate digit number and identity. Nature *418*, 979-983.

May, E.A., Kalocsay, M., D'Auriac, I.G., Schuster, P.S., Gygi, S.P., Nachury, M.V., and Mick, D.U. (2021). Time-resolved proteomics profiling of the ciliary Hedgehog response. J Cell Biol *220*.

Moore, B.S., Stepanchick, A.N., Tewson, P.H., Hartle, C.M., Zhang, J., Quinn, A.M., Hughes, T.E., and Mirshahi, T. (2016). Cilia have high cAMP levels that are inhibited by Sonic Hedgehog-regulated calcium dynamics. Proc Natl Acad Sci U S A *113*, 13069-13074.

Mukhopadhyay, S., Wen, X., Ratti, N., Loktev, A., Rangell, L., Scales, S.J., and Jackson, P.K. (2013). The ciliary G-protein-coupled receptor Gpr161 negatively regulates the Sonic hedgehog pathway via cAMP signaling. Cell *152*, 210-223.

Persson, M., Stamataki, D., te Welscher, P., Andersson, E., Bose, J., Ruther, U., Ericson, J., and Briscoe, J. (2002). Dorsal-ventral patterning of the spinal cord requires Gli3 transcriptional repressor activity. Genes Dev *16*, 2865-2878.

Shimada, I.S., Hwang, S.H., Somatilaka, B.N., Wang, X., Skowron, P., Kim, J., Kim, M., Shelton, J.M., Rajaram, V., Xuan, Z.*, et al.* (2018). Basal Suppression of the Sonic Hedgehog Pathway by the G-Protein-Coupled Receptor Gpr161 Restricts Medulloblastoma Pathogenesis. Cell reports *22*, 1169-1184.

Shimada, I.S., Somatilaka, B.N., Hwang, S.H., Anderson, A.G., Shelton, J.M., Rajaram, V., Konopka, G., and Mukhopadhyay, S. (2019). Derepression of sonic hedgehog signaling upon Gpr161 deletion unravels forebrain and ventricular abnormalities. Dev Biol *450*, 47-62.

Tschaikner, P.M., Regele, D., Rock, R., Salvenmoser, W., Meyer, D., Bouvier, M., Geley, S., Stefan, E., and Aanstad, P. (2021). Feedback control of the Gpr161-Galphas-PKA axis contributes to basal Hedgehog repression in zebrafish. Development *148*.